# Orchid: Flexible and Data-Dependent Convolution for Sequence Modeling

**Mahdi Karami**
Google Research
mahdika@google.com

**Ali Ghodsi**
School of Computer Science
University of Waterloo, ON, Canada
ali.ghodsi@uwaterloo.ca

## Abstract

In the rapidly evolving field of deep learning, the demand for models that are both expressive and computationally efficient has never been more critical. This paper introduces Orchid, a novel architecture designed to address the quadratic complexity of traditional attention mechanisms without compromising the ability to capture long-range dependencies and in-context learning. At the core of this architecture lies a new data-dependent global convolution layer, which contextually adapts its kernel conditioned on input sequence using a dedicated conditioning neural network. We design two simple conditioning networks that maintain shift equivariance in our data-dependent convolution operation. The dynamic nature of the proposed convolution kernel grants Orchid high expressivity while maintaining quasilinear scalability for long sequences. We evaluate the proposed model across multiple domains, including language modeling and image classification, to highlight its performance and generality. Our experiments demonstrate that this architecture not only outperforms traditional attention-based architectures such as BERT and Vision Transformers with smaller model sizes, but also extends the feasible sequence length beyond the limitations of the dense attention layers. This achievement represents a significant step towards more efficient and scalable deep learning models for sequence modeling.

## 1 Introduction

In modern deep neural networks, attention mechanisms have emerged as a gold standard, pivotal in domains such as natural language processing, image, and audio processing, and even complex fields like biology [Vaswani et al., 2017, Dosovitskiy et al., 2020, Dwivedi and Bresson, 2020]. However, despite their strong sequence analysis capabilities, these sequence modeling mechanisms suffer from their high computational complexity, which scales quadratically with sequence length, hindering their application to long-context tasks. This complexity has driven a shift towards innovative solutions to overcome this computational barrier, enabling analysis of long sequences in areas like genomics, DNA sequencing, and the creation of long musical compositions.

In the past years, researchers have explored various strategies to tackle the computational bottleneck of traditional dense attention layers [Tay et al., 2022]. One key strategy involves *sparsifying* the dense attention matrix. Instead of calculating the entire matrix, Qiu et al. [2019], Parmar et al. [2018] focus on specific local blocks of the receptive fields of sequences by chunking them into fixed-size blocks. Moreover, Sparse Transformer [Child et al., 2019], Longformer [Beltagy et al., 2020] and BigBird [Zaheer et al., 2020] use strided attention patterns combined with local sliding windows to reduce computation. In contrast to using pre-determined patterns, other techniques include learning to cluster/sort tokens based on a similarity function, thereby enhancing the global view of the sequence, as seen in Reformer [Kitaev et al., 2020], Routing Transformer [Roy et al., 2020] Sparse

Sinkhorn attention [Tay et al., 2020]. Another approach involves *low-rank approximations* of the self-attention matrix, leveraging the insight that these matrices often exhibit low-rank properties, as demonstrated by Linformer [Wang et al., 2020] which projects keys and values matrices to lower-dimensional representation matrices. Another paradigm to reduce quadratic computation cost, is to replace the dot-product similarity between keys and query matrices of attention mechanism with a *kernel function* and avoid explicitly computing the attention matrix [Katharopoulos et al., 2020]. Notable examples in this family include Performers [Choromanski et al., 2020], Random Feature Attention [Peng et al., 2021] that are based on random feature approximation of the kernel function. Additionally, some models leverage a combinations of such techniques to design an efficient transformer [Zhu et al., 2021, Zhang et al., 2021]. However, while these methods significantly reduce computational overhead, they may sacrifice expressiveness and performance, often requiring hybrid approaches that combine them with dense attention layers [Mehta et al., 2022, Fu et al., 2023]. On the other hand, recent works have aimed at sparsifying dense linear layers, used for feature mixing in Transformer blocks, to tackle another major source of high computation and memory demand in large models [Dao et al., 2022, Chen et al., 2021,].

Finding sub-quadratic and hardware-efficient mixing operators that are also expressive remains a significant challenge. Recent studies have explored attention-free solutions, particularly using state space models (SSMs) [Gu et al., 2021, Mehta et al., 2022, Wang et al., 2022, Fu et al., 2023, Orvieto et al., 2023, Gu and Dao, 2023, De et al., 2024], and long convolutions [Romero et al., 2021, Li et al., 2022, Poli et al., 2023]. A state space model characterizes a dynamical system's behavior in terms of its internal state using a state equation, describing the dynamics of the system using first-order differential equations over the states, and an observation equation, relating state variables to observed outputs.[1] A key insight is that, most of these SSM models can be formulated as a long convolution model between the input and output sequences [Gu et al., 2021], allowing parallel and efficient training. However, recent work by Poli et al. [2023] demonstrated that directly parameterizing the filter impulse response of the long-convolution leads to an even more expressive sequence mixing layer.

This paper proposes a novel data-dependent convolution mechanism to tackle the inherent quadratic complexity of traditional attention mechanisms, while maintaining the model's ability to capture long-range dependencies and in-context learning. The data-dependent convolution layer contextually adapts its kernel based on input data using a dedicated conditioning neural network. We design two simple yet effective conditioning networks that maintain shift equivariance in the adaptive convolution operation. By combining these adaptive mechanisms with gating operations, our proposed model—named *Orchid*—achieves high expressivity while offering quasilinear scalability (with a complexity of $\mathcal{O}(L \log L)$) for long sequences. Evaluation across various domains, including language modeling and image classification, presented in section 4 and Appendix, demonstrates the Orchid architecture's performance and generality, outperforming attention-based architectures, like BERT and Vision Transformers, with smaller model sizes. Moreover, its allows for handling very large sequence lengths that are beyond the limitations of the dense attention layers. This achievement lays the foundation for further advancements in more efficient and scalable sequence modeling architectures.

## 2  Background

**Self-Attention Mechanism:** Given a length-$L$ sequence of embeddings (tokens) $\boldsymbol{x} = (x_1, x_2, \ldots, x_L)$, the self-attention layer generates a new sequence by computing a weighted sum of these embeddings. To achieve this, it linearly project $\boldsymbol{x}$ into three components: queries ($\boldsymbol{Q}$), keys ($\boldsymbol{K}$), and values ($\boldsymbol{V}$), as: $\boldsymbol{Q} = \boldsymbol{x}\boldsymbol{W}^Q, \quad \boldsymbol{K} = \boldsymbol{x}\boldsymbol{W}^K, \quad \boldsymbol{V} = \boldsymbol{x}\boldsymbol{W}^V$. Each individual attention mechanism within a multi-head self-attention layer operates as a dense linear transformation, expressed as:

$$\boldsymbol{y} = \texttt{SA}(\boldsymbol{Q}, \boldsymbol{K}, \boldsymbol{V}) = \texttt{SoftMax}\left(\frac{\boldsymbol{Q}\boldsymbol{K}^T}{\sqrt{d_k}}\right)\boldsymbol{V} = \boldsymbol{A}(x)\boldsymbol{V},$$

where the matrix $\boldsymbol{A}(x)$ contains the normalized attention scores between each pair of tokens. This description of the attention layer highlights its notable benefits, including its capability to capture

---

[1]Notably, state space models, in general, include the recurrent layers such as RNN and LSTM [Hochreiter et al., 1997] as special cases.

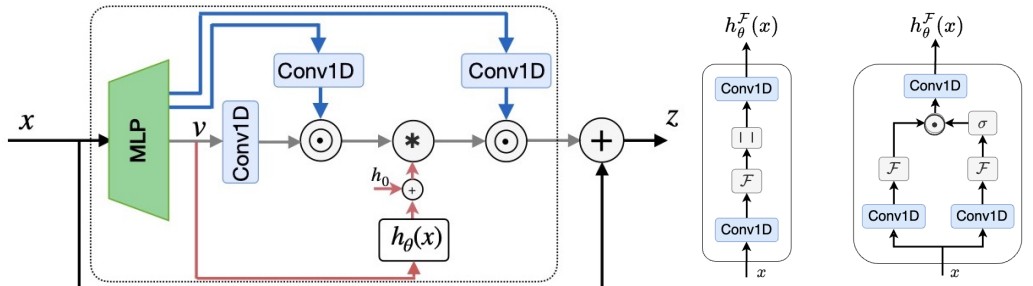

Figure 2.1: Orchid block architecture. This diagram illustrates the structure of the Orchid block. The core operation is a convolution (denoted by $*$), efficiently implemented in the frequency domain using FFT. Element-wise multiplication is denoted by $\odot$. On the right side, two different conditioning networks, introduced in equations (2) and (3) as shift-invariant convolution kernels, are illustrated. In this model, the convolution is performed efficiently in the spectral domain, so the kernel in the frequency domain, $h^{\mathcal{F}} = h_0^{\mathcal{F}} + h_\theta^{\mathcal{F}}(\boldsymbol{x})$, is computed. The block also includes MLPs for linear projection and pointwise mixing of features at the beginning, that is common design choice used in various sequence modeling architectures.

long-range dependencies using a sublinear parameter count. The attention mechanism enables direct computation of interactions between any two positions in the input sequence, regardless of their distance, without a corresponding rise in parameter counts. Additionally, the attention layer implements a *data-dependent* dense linear filter, which effectively filter the input based on on weights conditioned by a mapping of the data. This property makes it expressive and flexible enough to encode a large family of linear functions. However, these advantages come at the expense of quadratic computational complexity and memory costs.

This motivates us to develop an efficient and scalable *data-dependent convolution* mechanism, featuring an adaptive kernel that adjusts based on the input data. The kernel size of this convolution layer is as long as the input sequence length, enabling the capture of long-range dependencies across the input sequence while maintaining high scalability.

**Linear Convolution:** Discrete-time linear convolution is a fundamental operation in digital signal processing that calculates the output as the weighted sum of the finite-length input $\boldsymbol{x}$ with shifted versions of the convolution kernel, $\boldsymbol{h}$, also known as the impulse response of a linear time-invariant (LTI) system.[2] Formally, it can be written as

$$\boldsymbol{y}[t] = (\boldsymbol{h} * \boldsymbol{x})[t] \triangleq \sum_{\ell=0}^{L-1} h[t-\ell]x[\ell].$$

In this definition, the output is a linear filter of the *zero-padded* input and convolution kernel. However, other padding schemes leads to different forms of convolution. A well-known form is *circular convolution*, defined as

$$\boldsymbol{y}[t \bmod L] = (\boldsymbol{h} \circledast \boldsymbol{x})[t] \triangleq \sum_{\ell=0}^{L-1} h[t-\ell \bmod L]x[\ell],$$

which is equivalent to the linear convolution of two sequences if one is cyclically padded at its edges.

**Global Convolution and Fast Convolution Algorithm:** Standard convolution layers, explicitly parameterized with a short kernel, struggle to capture long-range dependencies in sequential data. Extending the kernel to match the input length enables modeling such dependencies but leads to linear growth in parameter counts and quadratic computational complexity. To mitigate the parameter growth challenge, the kernel of global (a.k.a. long) convolution can be implicitly

---

[2]While generally $\boldsymbol{x}$ represents a sequence of D-dimensional embeddings with length $L$, all the sequence mixing operators, including data-dependent convolutions, Fourier transforms and depthwise 1D convolutions (`Conv1d`) are performed along the sequence dimension. Therefore, for clarity and without loss of generality, we assume $D = 1$ and $\boldsymbol{x} \in \mathbb{R}^L$. For a full list of notation definition, please refer to Appendix A.

parameterized using a multilayer perceptron (MLP), a technique that has been shown to maintain sub-linear parameter scaling [Karami et al., 2019, Romero et al., 2021, Li et al., 2022, Poli et al., 2023]. Furthermore, a key advantage of convolution operators is that, leveraging the convolution theorem, convolution operators can be efficiently computed in the frequency domain using Fast Fourier Transform (FFT) algorithms, thereby reducing the computational complexity to $\mathcal{O}(L \log L)$ [Cooley and Tukey, 1965]. Formally, the linear convolution can be expressed in the frequency domain as $\hat{\boldsymbol{y}} = \mathcal{F}^{-1}(\mathcal{F}(\hat{\boldsymbol{h}}) \odot \mathcal{F}(\hat{\boldsymbol{x}})) = \boldsymbol{T}^{-1}(\boldsymbol{h}^{\mathcal{F}} \odot \boldsymbol{T}\hat{\boldsymbol{x}})$, where $\boldsymbol{T}$ is the DFT matrix, $\mathcal{F}$ denotes the discrete Fourier transformation, and $\hat{\boldsymbol{x}}$ denotes the zero-padded signal, defined as $\hat{\boldsymbol{x}} \triangleq \mathtt{pad}_{2L}(\boldsymbol{x}) = [\boldsymbol{0}_L; \ \boldsymbol{x}]$. Additionally, the circular convolution can be simply computed as: $\boldsymbol{y} = \boldsymbol{h} \circledast \boldsymbol{x} = \mathcal{F}^{-1}(\mathcal{F}(\boldsymbol{h}) \odot \mathcal{F}(\boldsymbol{x}))$.

# 3 Orchid Operator

This section introduces the *Data-Dependent Convolution Filter*, a novel operator aimed at increasing the expressiveness of long convolution operations. This operator serves as the foundational building block for the *Orchid* layer, which we will explore later in the section.

## 3.1 Data-Dependent Convolution Filter

We hypothesize that making the convolutional kernel data-dependent allows the filter to adapt to the specific characteristics of its input, potentially capturing more complex patterns within the sequence. Formally, this input-dependent filter is defined as:

$$\boldsymbol{y} = h_\theta(\boldsymbol{x}) * \boldsymbol{x} = \mathrm{NN}_\theta(\boldsymbol{x}) * \boldsymbol{x} \tag{1}$$

The key innovation is to replace the static convolutional kernel with a conditionally generated one controlled by the input data. This is achieved through a *conditioning network*, denoted as $h_\theta(\boldsymbol{x}) = \mathrm{NN}_\theta(\boldsymbol{x})$, a neural network parameterized by $\theta$. The conditioning network outputs a vector matching the input sequence in length. This allows each input token to 'attend' to the entire sequence with personalized, adaptive weights derived from its specific representation. Convolving the surrounding context using this data-dependent weighting scheme can potentially offer more effective sequence mixing compared to conventional static convolutions.

## 3.2 Preserving Shift-Equivariance in Data-Dependent Convolution

A fundamental property of discrete convolution is *shift equivariance*, meaning that shifting the input by a certain amount leads to a corresponding shift in the output (disregarding boundary effects). This is formally expressed for circular convolution as: $\mathtt{shift}_m(\boldsymbol{y}) = \boldsymbol{h} \circledast \mathtt{shift}_m(\boldsymbol{x})$ [Bronstein et al., 2021], where this property holds exactly regardless of boundary conditions. The shift operation is defined as $\mathtt{shift}_m(\boldsymbol{x})[t] \triangleq \boldsymbol{x}[t + m]$.

This property is particularly important because it ensures the operator's response is robust to shift of features within the input, thereby enhancing the model's generalization capabilities. This inductive bias is at the core of the widespread success of convolution operations [Thomas et al.]. Therefore, it is desirable to design conditioning network in the data-dependent convolution (1) to preserve shift equivariance property. To maintain this property for data-dependent convolution operations, it is sufficient to design filter kernel to be *shift-invariant*, *i.e.* $h(\mathtt{shift}_m(\boldsymbol{x})) = h(\boldsymbol{x})$ (refer to Appendix B for the proof). In the following, we present two conditioning network designs satisfying shift-invariance.

**I) Phase Suppression for Shift Invariance:** A circular shift of a sequence $\boldsymbol{u}$ results in a linear phase shift of its frequency components: $\mathcal{F}(\mathtt{shift}_m(\boldsymbol{u}))[\omega] = \boldsymbol{u}^{\mathcal{F}}[\omega] \cdot e^{-\frac{i2\pi}{L}\omega m}$ [Oppenheim, 1999]. Given a shift-equivariant function $g(x)$ (such as a depthwise $\mathtt{Conv1d()}$) (satisfying: $g(\mathtt{shift}_m(\boldsymbol{x})) = \mathtt{shift}_m(g(\boldsymbol{x}))$). Its frequency components after a spatial shift of its input maintain this phase shift:

$$\mathcal{F}(g(\mathtt{shift}_m(\boldsymbol{x}))[\omega] = \mathcal{F}(g(\boldsymbol{x}))[\omega] \cdot e^{-\frac{i2\pi}{L}\omega m}.$$

By taking the magnitude (absolute value or squared) of these complex-valued frequency components, we effectively eliminate the phase shift, therefore, defining $h^{\mathcal{F}}(\boldsymbol{x}) = \left|\mathcal{F}(g(\boldsymbol{x}))\right|$ satisfies shift-invariance property: $h^{\mathcal{F}}(\mathtt{shift}_m(\boldsymbol{x})) = h^{\mathcal{F}}(\boldsymbol{x})$.

In our design, we deploy a hybrid spatial-frequency domain conditioning network. This network consists of a 1D depthwise linear convolution (Conv1d()) with a short kernel length (typically 3-5) acting at the spatial domain, followed by a short convolution in the frequency domain. By operating in both spatial and frequency domains, the conditioning network effectively mixes information from neighboring tokens and spectral components. The resulting conditioning neural network is formulated as:

$$h_\theta^\mathcal{F}(\boldsymbol{x}) = \texttt{Conv1d}\big(\big|\mathcal{F}\left(\texttt{Conv1d}(\boldsymbol{x})\right)\big|\big) \tag{2}$$

This architecture choice aims to minimize the number of parameters and computational overhead introduced by the conditioning network within the overall model.

**II) Leveraging Cross-Correlation for Shift-Invariance:** An alternative approach to achieving shift-invariance involves computing the cross-correlation between two mapping of the input sequence. Let $k(\boldsymbol{x})$ and $q(\boldsymbol{x})$ be two shift-equivariant functions, satisfying: $k(\texttt{shift}_m(\boldsymbol{x})) = \texttt{shift}_m(k(\boldsymbol{x}))$ and $q(\texttt{shift}_m(\boldsymbol{x})) = \texttt{shift}_m(q(\boldsymbol{x}))$. We define $h(\boldsymbol{x})$ as the cross-correlation of $k(\boldsymbol{x})$ and $q(\boldsymbol{x})$, given by:

$$h(\boldsymbol{x})[t] = (k(\boldsymbol{x}) \star q(\boldsymbol{x}))[t] \triangleq \sum_{\ell=0}^{L-1} k(\boldsymbol{x})[\ell] \cdot q(\boldsymbol{x})[t + \ell \mod L].$$

This operation essentially slides $q(\boldsymbol{x})$ over $k(\boldsymbol{x})$ and measures their similarity at different offsets. Remarkably, the resulting cross-correlation function, $h(\boldsymbol{x})$, is also shift invariant:

$$\begin{aligned} h(\texttt{shift}_m(\boldsymbol{x})) &= k(\texttt{shift}_m(\boldsymbol{x})) \star q(\texttt{shift}_m(\boldsymbol{x})) \\ &= \texttt{shift}_m(k(\boldsymbol{x})) \star \texttt{shift}_m(q(\boldsymbol{x})) \\ &= k(\boldsymbol{x}) \star q(\boldsymbol{x}) = h(\boldsymbol{x}) \end{aligned}$$

Furthermore, the convolution theorem enables efficient computation of the cross-correlation in the frequency domain: $h^\mathcal{F}(\boldsymbol{x}) = \mathcal{F}\left(k(\boldsymbol{x}) \star q(\boldsymbol{x})\right) = k^{\mathcal{F}^*}(\boldsymbol{x}) \odot q^\mathcal{F}(\boldsymbol{x})$ where $k^{\mathcal{F}^*}$ denotes the complex conjugate of $k^\mathcal{F}$ and $\odot$ represents element-wise multiplication.

*Remark* 3.1. By setting $k(\boldsymbol{x}) = q(\boldsymbol{x}) = g(\boldsymbol{x})$, we obtain $h^\mathcal{F}(\boldsymbol{x}) = |g^\mathcal{F}(\boldsymbol{x})|^2$, This indicates that the cross-correlation approach generalizes the magnitude-based approach, demonstrating its versatility.

Similar to the previous approach, we employ separate 1D depth-wise short convolutions for both $k(\boldsymbol{x})$ and $q(\boldsymbol{x})$, followed by another convolution post cross-correlation in the frequency domain. As a result, the conditioning neural network is defined as

$$h_\theta^\mathcal{F}(\boldsymbol{x}) = \texttt{Conv1d}\Big(\mathcal{F}^*\left(\texttt{Conv1d}(\boldsymbol{x})\right) \odot \mathcal{F}\left(\texttt{Conv1d}(\boldsymbol{x})\right)\Big). \tag{3}$$

Both conditioning functions, as defined in (2) and (3), are illustrated schematically in Figure 2.1.

*Remark* 3.2. For convolution operations, we augment the data-dependent conditioning network by incorporating a fixed (static) term. This term adds positional encoding to the convolution kernel by implicitly parametrizing it using a positional embedding, PosEmb(), of time step (token index in the sequence) and a feed forward networks as $h_0 = \texttt{FFN}\big(\texttt{PosEmb}(t)\big)$ [Romero et al., 2021, Li et al., 2022, Poli et al., 2023]. The final convolution kernel is obtained by summing this positional bias with the output of the conditioning network, $h = h_\theta(\boldsymbol{x}) + h_0$.

*Remark* 3.3 (**Data-Dependent Convolution as a Cross-attention Alternative**). The kernel of convolution $h_\theta(\boldsymbol{x})$, defined in equations (2) or (3), is conditioned on the input of the convolution layer, making it input-dependent. However, we can generalize this concept further. The kernel could be a function of any arbitrary sequence $\boldsymbol{u}$, leading to a broader definition of data-dependent convolution:

$$\boldsymbol{y}(\boldsymbol{x}, \boldsymbol{u}) = h_\theta(\boldsymbol{u}) * \boldsymbol{x} = \text{NN}_\theta(\boldsymbol{u}) * \boldsymbol{x}$$

This definition couples the input sequence $\boldsymbol{x}$ with another sequence $\boldsymbol{u}$, creating a potential alternative to cross-attention layers in sequence processing tasks. We therefore refer to the proposed layer as *"data-dependent"* in a more general sense. When dealing with sequences of different lengths, the shorter sequence can be zero-padded to match the length of the longer one. Specifically, assuming $\boldsymbol{x} \in \mathbb{R}^L$ is longer than $\boldsymbol{u} \in \mathbb{R}^N$ ($L > N$), we use the zero-padded sequence $\hat{\boldsymbol{u}} = \texttt{pad}_L(\boldsymbol{u}) \in \mathbb{R}^L$ as input to the conditioning network $\text{NN}_\theta(\hat{\boldsymbol{u}})$. Since the long convolution is implemented in the frequency domain, this zero-padding in the time domain translates to interpolation in the frequency domain [Smith, 2008] ensuring that both sequences have frequency components of the same length.

### 3.3 Orchid Block

Unlike attention layers, convolution filters leverage parameter sharing. This means they slide the same kernel weights and apply them to different positions within the input sequence. Mathematically, this operation is equivalent to multiplying an input vector with a structured matrix, such as a Toeplitz matrix for linear convolutions or a circulant matrix for circular convolutions, which results in computational efficiency [Gray et al., 2006, Karami et al., 2019]. To achieve a location-dependent filtering scheme, we complement the data-dependent convolution with element-wise multiplications, allowing the model to emphasize specific tokens within by assigning higher weights prior to applying the location-invariant convolution. Notably, prior research has demonstrated that a cascade of circulant and diagonal matrices can effectively approximate dense linear layers [Moczulski et al., 2015, Cheng et al., 2015]. Building upon these insights, the overall architecture of the Orchid block, is composed of a chain of $M$ data-dependent convolution and element-wise multiplications (gated connections). In our experiments, we utilize a simple chain of order 1.5, consisting of data-dependent convolution sandwiched by two element-wise multiplications: $\boldsymbol{y} = (f_{\odot}^2 \circ f_* \circ f_{\odot}^1)(\boldsymbol{x})$ where $\circ$ denotes composition, $f_*(\boldsymbol{x}) \triangleq (h_\theta(\boldsymbol{x}) + h_0) * \boldsymbol{x}$, and $f_{\odot}^i(\boldsymbol{x}) \triangleq \texttt{Conv1d}(\boldsymbol{x}) \odot \boldsymbol{x}$. The Orchid block is illustrated in Figure 2.1 and its basic implementation is presented in appendix D.

**Overall Computational complexity.** All global convolutions within the Orchid block are computed in the frequency domain using FFT algorithm, inheriting its computational efficiency with complexity of $\mathcal{O}(L \log L)$. Furthermore, the element-wise multiplications contribute an additional $\mathcal{O}(L)$ complexity. Consequently, the overall complexity of the Orchid block scales quasi-linearly with the sequence length, resulting in a total complexity of $\mathcal{O}(ML \log L)$, where $M$ is the number of layers in the block. A recent hardware and I/O optimized implementation of FFT, introduced in [Fu et al., 2023], can speed up the overall computation of Orchid on modern accelerators. Empirical runtime comparisons against standard attention mechanisms, detailed in Appendix C.5, highlight its expected scalability, especially for longer sequences.

## 4 Experiments

Our evaluation of Orchid focuses on three different Transformer-based models to evaluate its expressivity and generalization capabilities as an alternative to attention layers. Firstly, we conduct a set of experiments on a synthetic task to assess the in-context learning ability and scalability of the proposed model. Subsequently, we evaluate the performance of the proposed architecture on language modeling tasks. Moreover, we extend our experiments to image classification tasks, aiming to evaluate the model's generalizability across diverse domains. Additional ablation studies on model architecture and also an experiments on raw speech classification with long sequences are presented in Appendices C.5 and C.2. Unless otherwise specified, our experiments adopt the phase supersession (Type I) conditioning network (Equation 2) due to its simpler form. For an overview experimental details, please refer to Appendix C.

### 4.1 Synthetic In-context Learning

The aim of the first experiment is to assess how well our model performs on a synthetic reasoning task. This task, inspired by prior work on language model benchmarking [Liang et al., 2022] and in-context learning (ICL) [Garg et al., 2022], is known as Associative Recall. It involves generating a value from a key given a string of key-value tuples from a random dictionary. For instance, given the input ([a, 1, b, e, f, 3], b), the model is expected to return e, the value associated with the key b. This task assesses whether a model can effectively retrieve the correct value from a key in a prompt, essentially applying a data-controlled shift. Attention mechanisms offer this capability by computing attention scores through token comparisons and then weighting the entire sequence accordingly [Olsson et al., 2022]. Associative recall has been pivotal in guiding the design of long convolution models, as demonstrated in [Fu et al., 2023], and a more complex variant of this task was employed in [Poli et al., 2023].

For these experiments, we benchmark Orchid against several leading long convolution models, including: I) H3, which utilizes state-space models (SSMs) for implicit parametrization of long convolution, as proposed in [Fu et al., 2023]. II) CKConv, that employs feedforward networks (FFNs) and positional embeddings for the implicit parametrization of convolution operations, detailed in [Romero et al., 2021]. III) Hyena, built upon the CKConv framework by incorporating an

Table 4.1: The performance (test accuracy) of in-context learning on the associative recall task with different sequence lengths and a vocabulary size of 20. The results for the baseline models are drawn from Poli et al. [2023], Fu et al. [2023]. The symbol ✗ indicates that the Transformer model failed to complete the task within a week or the model does not fit in memory.

| Model | 128 | 512 | 2K | 8K | 32K | 128K |
|---|---|---|---|---|---|---|
| Transformer | 100 | 100 | 100 | 100 | ✗ | ✗ |
| Monarch-Mixer | - | 98.7 | 99.4 | 99.4 | 99.4 | 99.4 |
| Hyena | 93 | 99 | 99.6 | 100 | 100 | - |
| Orchid | 100 | 100 | 100 | 100 | 100 | 100 |

Table 4.2: The test accuracy of the associative recall task with varying vocabulary sizes and a sequence length of 128.

| Model | 20 | 30 | 40 |
|---|---|---|---|
| Transformer | 100 | 100 | 100 |
| CKConv | 91 | 25.7 | 20.4 |
| H3 | 71.5 | 13.2 | 10.2 |
| Hyena | 93 | 38.8 | 12.4 |
| Mamba | 100 | 100 | 35.8 |
| Orchid | 100 | 99.4 | 99.2 |

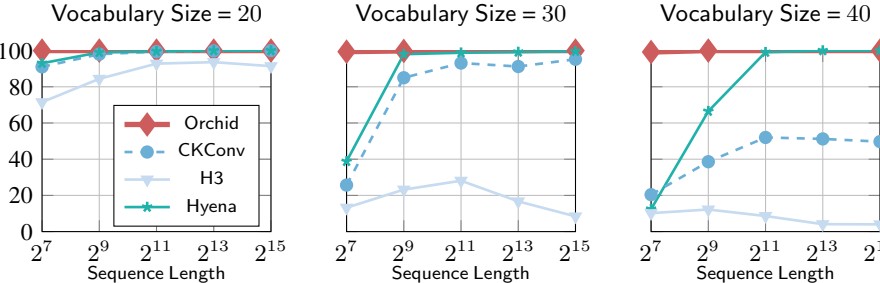

Figure 4.1: Test accuracy of the associative recall task across different long implicit convolution models on various sequence lengths and vocabulary sizes (number of possible token values).

additional exponential decay modulation into the implicit convolution process, as detailed in [Poli et al., 2023]. It is further augmented by a multiplication in a chain of order 2.

As illustrated in Figure 4.1 and Tables 4.1 and 4.2, Orchid demonstrates superior expressiveness and outperforms existing long convolution models in associative recall tasks. These tasks become increasingly challenging with shorter sequences and larger vocabulary sizes. This difficulty arises because specific (key, value) pairs appear less frequently within shorter strings, hindering the model's ability to learn and reason on these associations. Remarkably, in such challenging scenarios with short sequence lengths of 128 and large vocabulary sizes, Orchid significantly improves the model's accuracy and closes the gap between Transformer and implicit convolution models. Furthermore, Orchid successfully learns the task even with extended sequence lengths of up to 131K tokens, a scale at which Transformer models encounter computational difficulties, which highlights Orchid's superior scalability and efficiency in learning long context.

The insights from this experiment guide us in integrating the proposed model into Transformer-based models for extensive language modeling, suggesting its potential to enhance performance in natural language processing tasks.

## 4.2 Language Modeling

We evaluate the Orchid layer in language models. Orchid is designed to integrate seamlessly with existing BERT-style language models, such as BERT [Devlin et al., 2018], RoBERTa [Liu et al., 2019], SpanBERT [Joshi et al., 2020], and others [Jin et al., 2020]. In our experiments, we replace the attention layers in the standard BERT framework in [Devlin et al., 2018] with Orchid layers. For each Transformer block in the BERT-style model, we replace the attention layers with Orchid layers for sequence mixing. We also replace the two dense matrices in the MLP layers, used for dimension mixing in Transformers, with block-diagonal matrices [Dao et al., 2022]. Following [Fu et al., 2023], we add a residual long convolution to each Orchid layer.

Our BERT-style model, called Orchid-BERT-base, has 12 layers with a hidden size of 768, the same dimension and depth as BERT-base. The resulting Orchid-BERT-base have 77M parameters, compared to BERT-base's 110M parameters. We also pretrain Orchid-BERT-large of 254M parameters

Table 4.3: Average GLUE Score of BERT-base and BERT-large [Devlin et al., 2018] in comparison to Orchid-BERT-base and Orchid-BERT-base, and M2-BERT-base and M2-BERT-large Dao et al. [2022]. Baseline results are drawn from [Fu et al., 2023].

| Model (size) | GLUE Score | Δ Params | Δ GLUE Score |
|---|---|---|---|
| BERT-base (110M) | 79.6 | - | - |
| M2-BERT-base (80M) | 79.9 | -27.3% | +0.3 |
| Orchid-BERT-base (77M) | **80.6** | **-30.0%** | +1.0 |
| BERT-large (340M) | 82.1 | - | - |
| M2-BERT-large (260M) | 82.2 | -23.6% | +0.1 |
| Orchid-BERT-large (254M) | **82.7** | **-25.3%** | +0.6 |

with hidden dimensions of 1536 and 12 layers. Orchid models are pre-trained using masked language modeling over the C4 dataset [Raffel et al., 2019] with the `bert-base-uncased` tokenizer.

**Finetuning Performance on GLUE Benchmark.** We conducted an evaluation of Orchid-BERT models on GLUE fine-tuning tasks, comparing them against the baseline models: BERT-base and BERT-large, and the recent long convolution-based models: M2-BERT-base and M2-BERT-large [Fu et al., 2023]. The fine-tuning process was executed in accordance with the methodology described by Izsak et al. [2021]. As the results outlined in Table 4.3 show, Orchid-BERT-base is able to achieve 1.0 points improvement in average GLUE score performance compared to the BERT-base on the GLUE benchmark with utilizing 30% fewer parameters. Similarly, Orchid-BERT-large outperforms the performance of BERT-large by .6 points with a 25% reduction in parameter counts.

## 4.3 Image Classification

We extend the application of Orchid to the Vision Transformer (ViT) architecture, introduced by Dosovitskiy et al. [2020], by replacing its attention mechanism with Orchid similar to language modeling task. Similar to language modeling task, we substitute the dense matrices in the MLP layers, which perform dimension mixing, with block-diagonal matrices and incorporate a residual long convolution within each Orchid block. We benchmark our model against recent long convolution-based models, specifically Hyena-ViT-b [Poli et al., 2023] and M2-ViT-b [Fu et al., 2023].

Models are evaluated for image classification on two widely used image datasets: CIFAR-10 and ImageNet-1K. For CIFAR-10, images are transformed into sequences of $4 \times 4$ pixel patches and processed using a ViT architecture composed of 6 Transformer layers with hidden sizes of either 128 and 220 for Orchid-s and Orchid-m, respectively. In the case of ImageNet-1K, we segmented images into patches of $16 \times 16$ pixels, and we trained a ViT-base architecture featuring 12 Transformer layers and a hidden size of 768.

The results presented in Table 4.4 and 4.5 demonstrate that Orchid significantly outperforms both the Vision Transformer baseline and long convolution-based models on the CIFAR-10 and ImageNet-1K datasets. Notably, Table 4.5 shows that utilizing smaller image patches, which lead to longer sequences, can further enhance performance. This observation underscores the advantage of scalable sequence models like Orchid. These results confirm the generalizability and effectiveness of the Orchid architecture beyond the domain of language modeling, highlighting its potential advantage in broader range of applications such as image processing tasks.

## 5 Related Work

Previous works have explored various forms of dynamic convolution architectures [Wu et al., 2019, Karami et al., 2019, Chen et al., 2020, Jiang et al., 2020]. The dynamic convolution in [Wu et al., 2019] utilizes a short convolution kernel that depends solely on the current time-step, whereas [Jiang et al., 2020] expands the kernel span to depend on a local window. Meanwhile, Chen et al. [2020] modeled the convolution kernel as a mixture of short convolution kernels with mixture weights controlled by average pooling of the input embedding. However, the reliance on short convolutions and their specific kernel modeling approaches limits their ability to capture long-range dependencies and scaling their kernel to match the sequence length is computationally impractical. In a different line of research, Fourier Neural Operator (FNO) [Li et al., 2020] and adaptive FNO [Guibas et al.,

Table 4.4: Performance comparison of Orchid with ViT-based models on ImageNet-1k. Baseline results are drawn from [Fu et al., 2023].

| Model (size) | Top-1 (%) | Top-5 (%) |
|---|---|---|
| *ImageNet-1k* | | |
| ViT-b (87M) | 78.5 | 93.6 |
| ViT-b+Monarch (33M) | 78.9 | 94.2 |
| Hyena-ViT-b (88M) | 78.5 | 93.6 |
| M2-ViT-b (45M) | 79.5 | 94.5 |
| Orchid (48M) | **80.2** | **94.9** |

Table 4.5: Performance comparison of Orchid with ViT-based models on CIFAR-10 dataset. Orchid's performance is also evaluated over different patch sizes, $4 \times 4$, $2 \times 2$ and $1 \times 1$ pixels. Orchid-s and Orchid-m refers to the ViT architecture composed of 6 layers with hidden sizes of 128 and 220, respectively. Cross-Correlation (Type II) conditioning network (equation 3) is identified with -cc and the rest are using type I (equation 2). Baseline results are drawn from [Fu et al., 2023] and [Knigge et al., 2023].

| Model (size) | Top-1(%) | Model (size) | Top-1(%) | Model (size) | Top-1(%) |
|---|---|---|---|---|---|
| *CIFAR-10 $(4 \times 4)$* | | *CIFAR-10 $(1 \times 1)$* | | *CIFAR-10 $(2 \times 2)$* | |
| ViT (1.2M) | 78.6 | CKConv (1M) | 63.74 | Orchid-s (790K) | 92.2 |
| ViT+Monarch (607K) | 79.0 | S4 (7.8M) | 91.13 | Orchid-s-cc (799K) | 92.3 |
| Hyena-ViT (1.3M) | 80.6 | M2-ViT (741K) | 91.0 | Orchid-m (2.1M) | **93.33** |
| M2-ViT (741K) | 80.8 | CCNN (2M) | 93.08 | | |
| Orchid-s (735K) | **88.5** | Orchid-m (2M) | 93.0 | | |

2021] operate in the spectral domain, applying a dense linear layer or an MLP with a block diagonal linear layer. However, these models do not explicitly model convolution kernel and do not enforce shift-equivariance.

Recent advances in input-dependent state space models (SSMs) have shown promise in efficient sequence modeling by allowing model parameters to dynamically adapt based on the input [Gu and Dao, 2023]. However, the input-dependent mechanisms in these models typically rely on the current token or its local neighbors, preventing them from leveraging the benefits of global convolution for efficient parallelizable training Consequently, they heavily depend on hardware-aware implementations optimized for modern GPUs. While effective in certain tasks, these models have been shown to struggle with recall-intensive scenarios [Arora et al., 2024]. In contrast, our proposed data-dependent global convolution allows the conditioning network to be influenced by the entire input context, enabling efficient sequence mixing. Moreover, the current formulation of input-dependent SSMs is not readily adaptable for efficient cross-attention between sequence pairs. This highlights a promising future direction for research: exploring how the complementary strengths of data-dependent global convolutions and input-dependent SSMs can be combined to develop foundation models that excel across a broader range of tasks.

Recent studies have explored sub-quadratic sequence mixing methods using long convolutions or state space models, leveraging the fast convolution algorithm for computational efficiency. While utilizing Fast Fourier transform results in $\mathcal{O}(L \log L)$ computational complexity, FFT algorithms exhibit suboptimal hardware utilization and suffer from slow I/O between layers of the memory hierarchy on modern GPUs due to their sequential nature. To address this bottleneck, FlashFFT-Conv [Fu et al., 2023] utilizes a matrix decomposition to leverage matrix multiply units and enable kernel fusion resulting in a more hardware and I/O efficient implementation of long convolutions. Moreover, the Monarch Mixer (M2) [Fu et al., 2023], offers an expressive family of sub-quadratic structured matrices that generalizes the DFT and other structures. These matrices are parameterized as products of block-diagonal matrices, offering sub-quadratic computation costs ranging from $\mathcal{O}(L \log L)$ to $\mathcal{O}(L^{3/2})$. By trading-off computational complexity with FLOP utilization, M2 achieves a hardware-efficient alternative for Transformers.

# 6 Discussion and Conclusion

In conclusion, our work introduces Orchid, a novel model that addresses some critical challenges of efficiency and scalability in sequence modeling through the innovative use of data-dependent convolution. Orchid successfully mitigates the quadratic computational and memory costs associated with attention layers, while retaining, and in many cases enhancing, the model performance across different domains. The introduction of a data-dependent convolution layer represents a significant step forward, offering a scalable and expressive alternative sequence mixing scheme that adapts its weights to the input data. Through evaluation across multiple domains, including in-context learning, language and image processing tasks, Orchid has demonstrated not only its superior performance over traditional attention-based models but also its generality and scalability. This positions the Orchid model not just as a alternative to existing paradigms but as a potential catalyst for innovation, driving the exploration of novel, efficient, and powerful architectures in artificial intelligence.

The superior performance of Orchid compared to traditional transformer-based models raises intriguing questions about the current state and future direction of deep learning architectures. One plausible explanation for our model's effectiveness could be the over-parameterization prevalent in transformer-based models. A growing body of evidence indicates that attention mechanisms, despite their computational complexity, utilize only a fraction of their capabilities for tasks like language processing. This challenges the common belief that attention is the key ingredient for large-scale deep learning, leading us to reconsider its role and seek more computationally efficient alternatives.

**Limitations and Future Directions**

Looking ahead, extending our model to accommodate causal models, particularly for autoregressive language models akin to GPT, is an intriguing future direction. The current form of the proposed model is not inherently compatible with these architectures, primarily due to differences in how data-dependent global convolution handle dependencies and sequence generation.[3]

Furthermore, exploring the capability of Orchid as an efficient alternative to cross-attention layer, employed in sequence-to-sequence models, offers another avenue for research. These considerations open up new possibilities for integrating our model into more advanced foundation models and cross-domain applications.

Beyond sequence modeling, the Orchid block, with its input-dependent long convolution, local depthwise linear convolution (`Conv1d`), and element-wise multiplications, is inherently extendable to multi-dimensional data. While the primary focus of this work was designing an efficient and scalable architecture specifically for sequence modeling, expanding the proposed architecture to include 2D or 3D long convolutional approaches is an interesting future direction.

---

[3]Notably, recent works such as [Ding et al., 2023, Bachmann and Nagarajan, 2024] show that autoregressive models are not optimal for inference in language models.

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

## 7 Broader Impacts

The introduction of Orchid, with its innovative data-dependent convolution mechanism, can potentially lead to impacts across various sectors of society.

- **Accessibility and Democratization of AI**: By reducing the computational and memory requirements traditionally associated with deep learning models, Orchid makes advanced AI technologies more accessible to a broader range of researchers, startups, and institutions.

- **Environmental Sustainability**: The efficiency of Orchid, particularly in terms of computational and energy demands, aligns with the urgent need for environmentally sustainable AI practices. Lower energy consumption contributes to reducing the carbon footprint of training and deploying large-scale models, aligning with global sustainability goals.

- **Advancements in Healthcare**: In the healthcare sector, Orchid's ability to efficiently process long sequences of data could revolutionize early diagnosis and personalized medicine. For instance, it could enable more accurate analysis of genomic sequences or continuous health monitoring data.

# A  Notation definition

| Notations | Brief definition and interpretation |
|---|---|
| $\boldsymbol{x}, \boldsymbol{y}, \boldsymbol{W}$ | $\boldsymbol{x} \in \mathbb{R}^L$ and $\boldsymbol{y} \in \mathbb{R}^L$ are input and output sequence of a layer, while matrices are denoted by bold uppercase letters, such as layer weight matrix $\boldsymbol{W}$. |
| $\boldsymbol{h} * \boldsymbol{x}$ | linear convolution: $\boldsymbol{y}[t] = (\boldsymbol{h} * \boldsymbol{x})[t] \triangleq \sum_{\ell=0}^{L-1} h[t-\ell]x[\ell]$ |
| $\boldsymbol{h} \circledast \boldsymbol{x}$ | circular convolution: $\boldsymbol{y}[t \bmod L] = (\boldsymbol{h} \circledast \boldsymbol{x})[t] \triangleq \sum_{\ell=0}^{L-1} h[t-\ell \bmod L]x[\ell]$ |
| $\boldsymbol{k} \star \boldsymbol{q}$ | cross-correlation: $\boldsymbol{h}[t] = (\boldsymbol{k} \star \boldsymbol{q})[t] \triangleq \sum_{\ell=0}^{L-1} \boldsymbol{k}[\ell] \cdot \boldsymbol{q}[t+\ell \ \bmod L]$ |
| $\boldsymbol{h} \odot \boldsymbol{x}$ | element-wise multiplication (Hadamard product): $\boldsymbol{y}[t] = (\boldsymbol{h} \odot \boldsymbol{x})[t] \triangleq h[t] \cdot x[t]$ |
| $\texttt{shift}_m(\boldsymbol{x})$ | $\texttt{shift}_m(\boldsymbol{x})[t] \triangleq \boldsymbol{x}[t+m]$ |
| $\mathcal{F}(), \mathcal{F}^{-1}(), \boldsymbol{T}$ | The forward and inverse discrete Fourier transforms while $\boldsymbol{T}$ represents the DFT matrix. |
| $\boldsymbol{x}^{\mathcal{F}}$ | $\boldsymbol{x}^{\mathcal{F}} = \mathcal{F}(\boldsymbol{x}) = \boldsymbol{T}\boldsymbol{x}$ |
| $\hat{\boldsymbol{x}}$ | $\hat{\boldsymbol{x}} \in \mathbb{R}^{2L}$ is a zero-padded signal at the left side defined as $\hat{\boldsymbol{x}} \triangleq \texttt{pad}_{2L}(\boldsymbol{x}) = [\boldsymbol{0}_L; \ \boldsymbol{x}]$ |
| $\texttt{pad}_N(\boldsymbol{x})$ | padding the vector $\boldsymbol{x} \in \mathbb{R}^L$ with zeros to the target size $N$, *i.e.* $\texttt{pad}_N(\boldsymbol{x}) = [\boldsymbol{0}_{N-L}; \ \boldsymbol{x}]$ where $[\boldsymbol{0}_{N-L}]$: a zero vector of size $(N-L)$. |
| $h_\theta(\boldsymbol{x})$ | *conditioning network* (data-dependent convolution kernel): $h_\theta(\boldsymbol{x}) = \text{NN}_\theta(\boldsymbol{x})$ that is a neural network parameterized by $\theta$. |
| $\texttt{Conv1d}$ | 1D depthwise linear convolution with a short kernel length (typically 3-5) applied to each feature dimension |
| $\texttt{FC}()$ | dense fully connected layer $\texttt{FC}(\boldsymbol{x}) = \boldsymbol{x}\boldsymbol{W} + \boldsymbol{b}$ |
| $\texttt{MLP}()$ | Multi-Layer Perceptron which is composed of dense fully connected layers |
| $\texttt{PosEmb}()$ | a positional embedding of time step $t$ |
| $h(t)$ | positional encoding part of the convolution kernel: $h(t) = \texttt{FFN}\big(\texttt{PosEmb}(t)\big)$ |

*Remark* A.1 (**Orchid Combines Global and Local Mixing Operations in Spatial and Spectral Domains:**). The core of Orchid is a long, data-dependent convolution with an adaptive kernel spanning the entire input sequence. This means that the convolution operation performed by Orchid mixes (convolves) the entire sequence globally using the proposed adaptive input dependent kernels. On the other hand, the conditioning networks responsible for generating this adaptive kernel operate locally in both the spectral and spatial domains.

Comparing self-attention to Orchid reveals further insights. The pair of mappings $k(x)$ and $q(x)$ in the conditioning network of Orchid (equation 3), are analogous to the key $k(x)$ and query $q(x)$ in the attention mechanism. In both models, these components are modeled using local neural networks: attention utilizes pointwise linear projections, while Orchid employs local $Conv1D$ operations in both the spatial and spectral domains. The attention mechanisms calculate the input-dependent attention score matrix $A(x)$ and subsequently compute the output as $y = A(x)v$. In contrast, Orchid 's conditioning network performs a cross-correlation— a global operation—between $k(x)$ and $q(x)$ to derive the convolution kernel $h(x)$, followed by global convolution $y = h(x) * v$. Therefore, while the inner blocks (the input-dependent networks that compute the convolution kernel in Orchid) operate locally on the inputs, the outer blocks (such as the matrix product in attention, cross-correlation or convolution in Orchid) perform global sequence mixing. This approach ensures fixed (or sublinear) parameter scaling with respect to sequence length, preventing the model size from growing excessively with sequence length.

*Remark* A.2 (**Model Name:**). The name "*Orchid*" for our model is more than just a label; it carries a symbolic meaning for our model, reflecting its elegance, resilience, and adaptability. Orchids are known to thrive in diverse environments and exhibit subtle color variations under specific environmental conditions, including light intensity, seasonal changes, and dyeing. The essence of adaptation and efficient resource utilization resonates profoundly with our model's design. Moreover, the proposed model's computational efficiency aligns with more environmentally sustainable AI practices by minimizing energy consumption and carbon footprint during training and deployment.

# B  Proof

**Proof for Preserving Shift Equivariance:**  Given a *circular shift invariant* filter kernel, where

$$h_\theta(\boldsymbol{x}[t + m \text{ mod } L]) = h_\theta(\boldsymbol{x}[t]).$$

For a circularly shifted input, the circular convolution can be expressed as:

$$
\begin{aligned}
(\boldsymbol{h}_\theta(\texttt{shift}_m(\boldsymbol{x})) \circledast \texttt{shift}_m(\boldsymbol{x}))[t] &= \sum_{\ell=0}^{L-1} h_\theta(\texttt{shift}_m(\boldsymbol{x})[\ell]) \cdot \texttt{shift}_m(\boldsymbol{x})[t - \ell \text{ mod } L] \\
&= \sum_{\ell=0}^{L-1} h_\theta(\boldsymbol{x}[\ell + m]) \cdot \texttt{shift}_m(\boldsymbol{x})[t + m - \ell \text{ mod } L] \\
&= \sum_{\ell=0}^{L-1} h_\theta(\boldsymbol{x}[\ell]) \cdot \texttt{shift}_m(\boldsymbol{x})[t + m - \ell \text{ mod } L] \\
&= \boldsymbol{y}[t + m \text{ mod } L] \\
&= \texttt{shift}_m(\boldsymbol{y})[t]
\end{aligned}
$$

Therefore, if the filter kernel is shift-invariant, the conditional convolution will also be shift-equivariant. ∎

# C  Experimental Details

## C.1  Synthetic In-context Learning

We used associative recall task to evaluate in-context learning of the proposed model. Given a string of key-value tuples from a random dictionary, this task involves generating a value from a key . For instance, given the input ($[a, 1, b, e, 3, f]$, $b$), the model is expected to return $e$, the value associated with the key $b$. This task assesses whether a model can effectively retrieve the correct value from a key in a prompt, essentially applying a data-controlled shift. While attention mechanisms offer this capability by computing pair-wise attention scores and then weighting the entire sequence accordingly [47], they face computational difficulties for very large sequence.

The model is composed of 2 Orchid block while each Orchid is composed of a chain of order $1.5$, including 2 element-wise multiplications and one data-dependent convolution. The hidden size of all models in this experiments are set to $64$. The conditioning network, defined in (2), was used which was composed of a 1D depthwise convolution (`Conv1d`) in the time domain and a `Conv1d` in the frequency domain that were applied separately to each feature dimension, both having a kernel of length 3. Only for vocabulary size of 40 with short sequence lengths of 128 we increased the depth to 3 layers of `Conv1d` in the spatial domain and in the frequency domain.

For training, we used Adam optimizer [63] with its standard settings ($\beta_1 = .9, \beta_2 = .999$), and learning rate of 5e-4 with linear warmup schedule in 1000 steps. A weight decay of $0.1$ was used as a regularizer. The Orchid models were trained Orchid on a single P100 GPU for small to medium sequence length and on a single V100 GPU for long sequences.

## C.2  Model Architecture Ablation

To better understand the impact of different model components, we conducted several ablation studies on in-context learning task in section 4.1. First, we evaluated different depthwise linear convolution architectures in the conditioning network, as outlined in Equation (2). This ablation study compared: I) *Spatial then Spectral:* applying `Conv1D` in the spatial domain followed by `Conv1D` in the frequency domain (as proposed in Equation (2)), II) *Spatial Only:* applying two `Conv1D` layers in the spatial domain only, and III) *Spectral Only:* applying two `Conv1D` layers in the spectral domain only. The results, presented in Figure C.1, demonstrate that the proposed method of operating in both spatial and frequency domains (Equation 2) offers the best performance. This approach effectively mixes information from neighboring tokens in both domains, capturing richer representations.

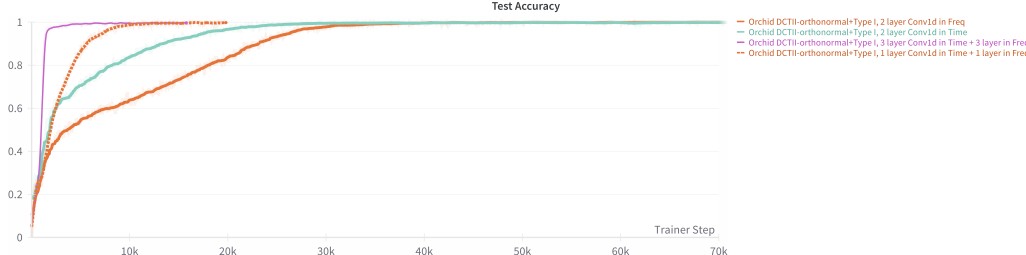

Figure C.1: Comparison of Local `Conv1D` Choices: Evaluation of different local convolution options used in the conditioning network. Conditioning networks of type I (Equation 2) (1 layer `Conv1D` in time + 1 layer in frequency), 2 layer `Conv1D` in time, 2 layer `Conv1D` in frequency, and 3 layer `Conv1D` in time + 3 layer in frequency.

Second, we compared the two conditioning networks for data-dependent convolution defined in Equations 2 and 3. We also evaluated various nonlinearity $\sigma()$ functions used in the Type II (cross-correlation) conditioning network:

$$h_\theta^{\mathcal{F}}(\boldsymbol{x}) = \texttt{Conv1d}\Big( \mathcal{F}^* \left( \texttt{Conv1d}(\boldsymbol{x}) \right) \odot \sigma\big( \mathcal{F} \left( \texttt{Conv1d}(\boldsymbol{x}) \right) \big) \Big). \tag{4}$$

The nonlinearities includes $\{\texttt{Tanh()}, \texttt{Sigmoid()}, \texttt{Softsign()}, \texttt{Softshrink()}, \texttt{Identity()}\}$, each applied only to the magnitude of its argument. The results in Figure C.2 indicates that removing this nonlinearity provides the best performance, slightly exceeding that of `Softshrink()` and `Tanh()`. Notably, among the nonlinearities those that cross zero perform better. Moreover, we observed that Type II conditioning networks (cross-correlation) with `Identity()` and `Softshrink()` exhibit faster convergence compared to Type I (phase-suppression).

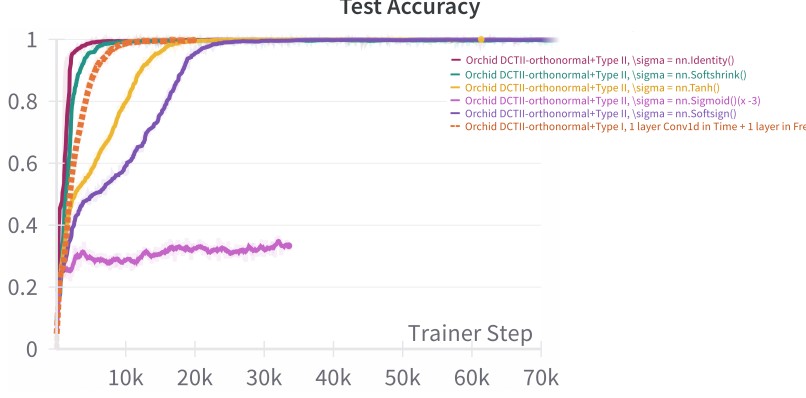

Figure C.2: Comparison of different $\sigma()$ on conditioning network of Type II (cross-correlation in equation 4).

Finally, we evaluated two different Fourier transforms: Discrete Fourier Transform (DFT) and Discrete Cosine Transform (DCT). We considered both orthogonal and orthonormal versions of each transform. Orthonormal transforms utilize an orthogonal and normalized basis, resulting in a unitary transform matrix $\boldsymbol{T}$. The results, presented in Figure C.3, show that Type I conditioning networks (Equation 2) exhibit faster convergence. Moreover, the DCT surpasses the DFT in terms of learning speed, likely because DCT corresponds to even-symmetric padding, which smooths the signal at the boundaries compared to the circular padding of the DFT. Furthermore, the DCT produces real-valued transforms. Notably, using orthonormal transforms accelerates the learning process. Consequently, we opted for the orthonormal DCT with Type I conditioning networks for data-dependent convolution in all the experiments.

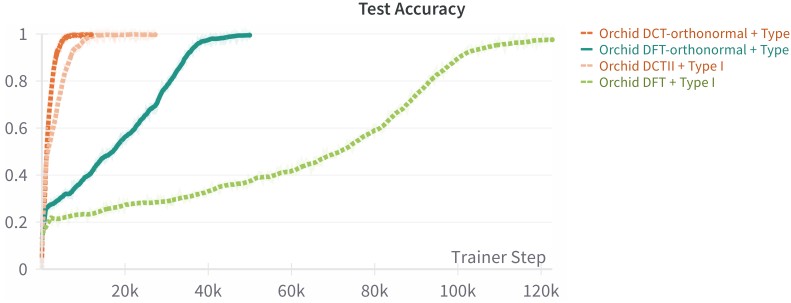

Figure C.3: Test accuracy of in-context learning on the associative recall task with a vocabulary size of 20 and sequence length of 128, comparing different model components. `Type I` refers to conditioning networks of type I (based on absolute value in Equation 2). `Orthonormal` indicates transforms that utilize orthogonal and normalized bases.

## C.3 Language Modeling

In our experiments, we replaced the attention layers in the standard BERT framework in [48] with Orchid layers. For each Transformer block in the BERT-style model, we replaced the attention layers with Orchid layers for sequence mixing. Each Orchid layer was composed of a chain of order $1.5$, including 2 element-wise multiplications and one data-dependent convolution. The conditioning network, defined in (2), was used which was composed of a short `Conv1d` in time domain and a `Conv1d` in the frequency domain that were applied separately to each feature dimension, both having a kernel of length 3. For dimension mixing, we also substituted the two dense matrices in the MLP layers with block-diagonal matrices of order 1 with $b = 4$ blocks [21] and also add a residual long convolution to each Orchid layer, as in [44].

Our BERT-style model, called Orchid-BERT-base, was composed of 12 layers with a hidden size of 768, the same dimension and depth as BERT-base resulting in Orchid-BERT-base having 77M parameters, compared to BERT-base's 110M parameters. We also pre-trained Orchid-BERT-large of 254M parameters with hidden dimensions of 1536 and 12 layers.

For pre-training, we used decoupled Adam optimizer with $\beta_1 = .9, \beta_2 = .98$, and learning rate of 8e-4 with linear warmup schedule within first 6% of the steps and then decay with linear rate. A weight decay of 1e-5 is used as a regularizer. Orchid models were pre-trained using masked language modeling with 30% masking over the C4 dataset [52] with sequence length of 128 and the `bert-base-uncased` tokenizer. Models were pre-trained on a node of 4xA100 GPUs for 70k steps with batch size of 4096.

**Finetuning on GLUE Benchmark:** To evaluate the performance of Orchid-BERT models, we finetuned the pre-trained models on GLUE fine-tuning tasks, comparing them against the baseline models: BERT-base and BERT-large, and the recent long convolution-based models: M2-BERT-base and M2-BERT-large [44]. The fine-tuning process was executed in accordance with the methodology described by Izsak et al. [53].

For all tasks, the sequence length was 128. For some of the tasks, we applied average pooling on the embeddings of all the non-padding tokens and use it as the model output. We followed [53] in fine-tuning small dataset tasks: RTE, MRPC, and STS-B are initialized from the fine-tuned checkpoint of MNLI dataset. Models were fine-tuned on a node of 4xA100 GPUs.

In contrast to [44], which performed a hyperparameter search for learning rate, weight decay, and number of epochs, we used the reported hyper-parameters in [44] and didn't perform a thorough search for them. As noted in [53], hyperparameter search can result in substantial improvements in the performance. In particular for CoLA task, whose results are bellow M2-BERT, we anticipate that fine tuning the parameter will improve its performance. The detailed hyperparameters for each task is reported in Table C.2.

Table C.1: A summary of the Fine-tuning results on GLUE benchmark [64]. Following the the methodology outlined in [53], we use the standard evaluation metrics: Spearman's correlation for STS-B, Matthew's correlation for CoLA, F1 scores for QQP and MRPC, and accuracy for the other tasks. For MNLI, the average of multiclass accuracy (m) both multiclass accuracy of mismatched set (mm) is used.

| Model | MNLI (m / mm) | RTE | QNLI | QQP | SST2 | STS-B | CoLA | MRPC | **Average** |
|---|---|---|---|---|---|---|---|---|---|
| M2-BERT-base (80M) | 78.4 / 78.6 | 68.5 | 84.6 | 86.7 | 92.0 | 86.3 | 53.0 | 89.8 | 79.9 |
| Orchid-BERT-base (77M) | 79.53 / 80.52 | 70.4 | 86.16 | 87.2 | 92.05 | 86.63 | 51.76 | 90.53 | **80.6** |
| M2-BERT-large (260M) | 81.7 / 81.9 | 72.8 | 84.7 | 87.8 | 93.3 | 88.0 | 59.2 | 90.0 | 82.2 |
| Orchid-BERT-large (254M) | 81.82 / 82.74 | 76.32 | 87.02 | 88.18 | 92.8 | 88.29 | 56.97 | 89.24 | **82.65** |

Table C.2: Hyperparameters for Orchid-bert-base (77M) and Orchid-BERT-large (254M) on Various GLUE Tasks. D-AdamW stands for Decoupled AdamW optimization algorithm. For QNLI task in Orchid-bert-base (77M) we applied everage pooling.

| Hyperparameter | MNLI | RTE | QNLI | QQP | SST2 | STSB | CoLA | MRPC |
|---|---|---|---|---|---|---|---|---|
| **Orchid-bert-base (77M)** | | | | | | | | |
| Optimizer | D-AdamW | AdamW | AdamW | AdamW | D-AdamW | AdamW | D-AdamW | AdamW |
| Learning Rate | 5e-5 | 5e-5 | 5e-5 | 3e-5 | 3e-5 | 7e-5 | 5e-5 | 5e-5 |
| Weight Decay | 5e-6 | 0.01 | 1e-5 | 0.01 | 3e-6 | 0.01 | 5e-6 | 0.01 |
| Epochs | 3 | 6 | 10 | 10 | 3 | 10 | 10 | 10 |
| **Orchid-bert-large (254M)** | | | | | | | | |
| Optimizer | D-AdamW | D-AdamW | D-AdamW | D-AdamW | D-AdamW | D-AdamW | AdamW | D-AdamW |
| Learning Rate | 5e-5 | 1e-5 | 5e-5 | 3e-5 | 3e-5 | 7e-5 | 5e-5 | 8e-5 |
| Weight Decay | 5e-6 | 1e-6 | 1e-5 | 3e-6 | 3e-6 | 3e-6 | 5e-6 | 8e-6 |
| Epochs | 3 | 3 | 10 | 5 | 3 | 10 | 10 | 10 |

## C.4 Image Classification

We extended the application of Orchid to the Vision Transformer (ViT) architecture, introduced by Dosovitskiy et al. [2], by replacing its attention mechanism with Orchid similar to language modeling task. The sinusoidal positional embeddings and global average-pooling were employed for the class token, following [65, 66]. We compared the performance of our model against recent long convolution-based models, specifically Hyena-ViT-b [31] and M2-ViT-b [44]. We evaluated the models on two widely used image classification datasets: CIFAR-10 and ImageNet-1K.

Each Orchid layer consists of a chain of order 1.5, involving 2 element-wise multiplications and one data-dependent convolution. The conditioning network (defined in Eq. 2) employs a combination of `Conv1d` layers in both the time and frequency domains, applied separately to each feature dimension with a kernel length of 3 and 5 for CIFAR-10 and ImageNet-1K, respectively.

**CIFAR-10:** For CIFAR-10, images were transformed into sequences of $4 \times 4$ pixel patches and processed using a ViT architecture composed of 6 Transformer layers with hidden sizes of either 128 and 220 for Orchid-s and Orchid-m, respectively. For training, we used Adam optimizer with its standard setting ($\beta_1 = .9, \beta_2 = .999$), and learning rate of 3e-4 with linear warmup schedule within first 500 steps. For the experiments on Orchid-s ($4 \times 4$), we tuned base learning rate over (1e-3, 5e-3, 1e-2). A weight decay of 0.05 was used as a regularizer. Orchid was trained on a single P100 GPU for 500 epochs with batch size of 512.

**ImageNet-1K:** We segmented images into patches of $16 \times 16$ pixels, and we trained a ViT-base architecture featuring 12 Transformer layers and a hidden size of 768. We incorporated a residual long convolution within each Orchid layer and substituted the dense matrices in the MLP layers responsible for feature mixing with block-diagonal matrices of order 1 with $b = 4$ blocks, reducing their computational complexity and the total model's size.

For training, we used Adam optimizer with its standard setting ($\beta_1 = .9, \beta_2 = .999$), and base learning rate of 1e-3 with linear warmup schedule within first 10 epochs and then decay with Cosine schedule. For data augmentation, we adopted the T2T-ViT pipeline [67], including Random erasing with rate=0.25, CutMix with $\alpha$=1.0, Mixup with $\alpha$=0.8, AugMix, and RandAugment with [magnitude=9, magnitude-std=0.5, layers=2]. We trained Orchid on 4xA100 GPUs for 300 epochs and batch size of 1024.

Table C.3 summarizes architecture and training settings.

Table C.3: Architecture and training settings for Orchid models.

|  | ImageNet-1k | CIFAR-10 |
| --- | --- | --- |
| Number of Orchid blocks | 12 | 6 |
| Hidden dimension | 768 | 128, 220 |
| Short `Conv1d` kernel size | 5 | 3 |
| `PosEmb` dimension | 33 | 5 |
| Patch size | $16 \times 16$ | $4 \times 4$ |
| Batch size | 1024 | 512 |
| Training epochs | 300 | 500 |
| Weight decay | 0.05 | 0.05 |
| Learning rate schedule | Cosine decay with linear warmup | Constant with linear warmup |
| Base learning rate | 1e-3 | (1e-3, 5e-3, 1e-2) |
| Warmup | 10 epochs | 500 steps |
| Label smoothing | 0.1 | 0.1 |

Table C.4: Accuracy on the Speech Commands dataset with 10 classes. S4-M2 refers to the S4 model [24] where dense matrices in the MLP layers are replaced with block-diagonal matrices. Baseline results are sourced from [44], and ✗ indicates that the Transformer model could not fit in GPU memory.

| Transformer | Performer | CKConv | WaveGan-D | S4 | S4-M2 | Orchid |
| --- | --- | --- | --- | --- | --- | --- |
| ✗ | 30.8 | 71.7 | 96.3 | 97.5 | **97.9** | 97.7 |

## C.5  Speech Classification

To evaluate the ability to learn long-range dependencies in real-world time-series data, we conducted experiments on the speech classification task using the SC10 subset of the Speech Commands dataset, which contains 10 classes. Following [29], we use the raw speech dataset with 1-second duration sampled at 16000 Hz for this task. The performance results shown in Table C.4 highlight that Orchid performs on par with the state-of-the-art model on this long sequence modeling task.

## C.6  Runtime Benchmark

To assess the computational efficiency of Orchid, we benchmarked its runtime against both traditional attention layers and the optimized FlashAttention mechanism [68].The evaluation was conducted on an NVIDIA `A100-40GB` GPU with models featuring a hidden dimension of 768 and a batch size of 1. The empirical runtime comparisons in Figure C.4 highlights Orchid's high scalability, especially for longer sequences. Here, FlashAttention utilizes `float16` precision, while Attention and Orchid use `float32`. It's worth noting that Orchid's convolution operators utilize the standard FFT implementation in PyTorch, and further speed improvements are anticipated with the integration of fused CUDA kernels or FlashFFT [43].

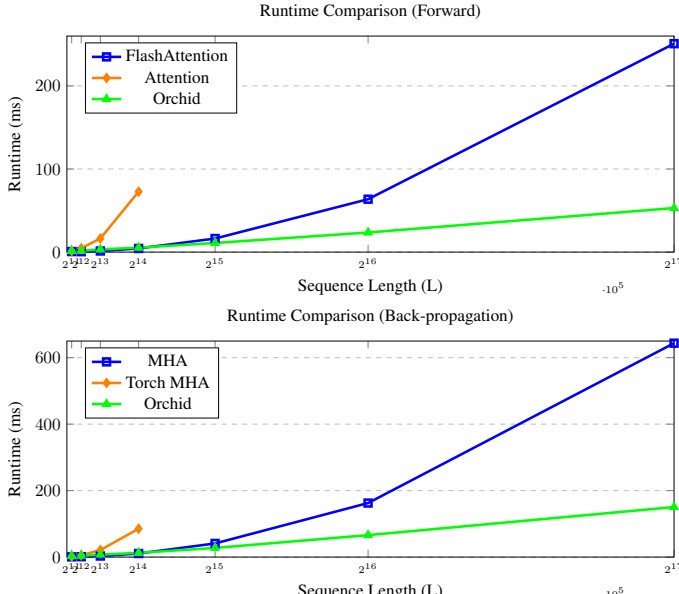

Figure C.4: Forward and backward runtime comparison of different attention mechanisms (FlashAttention, Attention, and Orchid) with varying sequence lengths.

## D   Implementation

```python
1  import torch
2  import torch.nn as nn
3  from torch_dct import dct, idct
4  from einops import rearrange
5
6  class DiscreteTransform(nn.Module):
7    def __init__(self, mode="dct", norm="ortho", dim=1):
8      super().__init__()
9      self.dim = dim; self.mode = mode; self.norm = norm
10
11   def forward(self, x):
12     if self.mode == "dct":
13       return dct(x, dim=self.dim, n=x.shape[self.dim])
14       # return rearrange( dct(rearrange(x,"b l d -> b d l"), norm=self.norm), "b d l -> b
15     elif self.mode == "fft":
16       return torch.fft.rfft(x, dim=self.dim)
17
18   def inverse(self, x):
19     if self.mode == "dct":
20       return idct(x, dim=self.dim, n=x.shape[self.dim])
21     elif self.mode == "fft":
22       return torch.fft.irfft(x, dim=self.dim)
23
24
25 class OrchidOperator(nn.Module):
26   def __init__(self, d, L, d_filter=64, l_conv1d=3, dxt_mode="fft", to_out_proj=True):
27     super().__init__()
28
29     self.d_model = d
30     self.seq_len = L
31
32     # setup projections
33     self.in_linear = nn.Linear(d, 3*d)
34     if to_out_proj:
35       self.out_linear = nn.Linear(d, d)
36
37     # setup short conv filter
38     width = d * 2
39     self.short_filter = nn.Conv1d(width, width,
40         kernel_size=l_conv1d, groups=width, padding=l_conv1d-1)
41
42     # setup static conv
43     """ The implementation of static conv in Hyena (inspired by CKConv) was used.
44         A basic implementation is:
45         nn.Sequential(PositionalEmbedding(emb_dim, l),
46             nn.Linear(emb_dim,d_filter), Sin(), nn.Linear(d_filter,d_filter),
47             ExponentialModulation())"""
48     self.static_conv = CKConv(d, order=d_filter, seq_len=L)
49
50     # Setup conditioning network. See shift invariant model: I
51     self.conditioning_nn = nn.Sequential(
52       nn.conv1d(d, d, l_conv1d, d, padding=l_conv1d-1),
53       DiscreteTransform(mode=dxt_mode, dim=1),
54       nn.abs(),
55       nn.conv1d(d, d, l_conv1d, d, padding=l_conv1d-1),
56     )
57
58     self.transform = DiscreteTransform(mode=dxt_mode, dim=-1)
59
60   def forward(self, x, **kwargs): # x is (b, l, d)
61     x = self.in_linear(x)
62     _, _, v = torch.split(x, self.d_model, dim=-1)
63     h_adapt_f = self.conv_kernel_nn(v) # Input-dependent Kernel
```

```
64    h_0 = self.static_conv.filter(self.seq_len)
65    h_0 = rearrange(h_0, "c l d -> c d l")
66
67    # short conv1d() filter
68    x = rearrange(x, "b l d -> b d l")
69    x = self.short_filter(x)[..., :self.seq_len]
70
71    s1, s2, v = x.split(self.d_model, dim=1)
72
73    y = v * s1
74    y = self.adaptive_conv(y, h_0=h_0, h_adapt_f=h_adapt_f)
75    y = y * s2
76    y = rearrange(y,"b d l -> b l d")
77    if self.to_out_proj:
78      y = self.out_linear(y)
79    return y
80
81 def adaptive_conv(self, x, h_0, h_adapt_f): # x is (b, d, l)
82    h_0_f = self.transform(h_0)
83    x_f = self.transform(x)
84    y = torch.fft.irfft(x_f * (h_0_f + h_adapt_f))
85    return y[..., :self.seqlen]
```

Listing 1: A basic implementation of the Orchid layer.

