# OpenReview forum: "Orchid: Flexible and Data-Dependent Convolution for Sequence Modeling"
_NeurIPS.cc/2024/Conference — NeurIPS 2024 poster_

### Official Review · Reviewer_eXPv · 2024-07-06

**Soundness:** 3
**Presentation:** 3
**Contribution:** 3
**Rating:** 7
**Confidence:** 4

**Summary:**

This paper introduces Orchid, a novel deep learning architecture that addresses the quadratic complexity of traditional attention mechanisms while still capturing long-range dependencies and enabling in-context learning. The key innovation is a data-dependent global convolution layer that dynamically adapts its kernel based on the input sequence using a dedicated conditioning neural network. Two simple conditioning networks are designed to maintain shift equivariance in the data-dependent convolution operation. The dynamic convolution kernel allows Orchid to achieve high expressivity with quasilinear scalability for long sequences. Experiments across language modeling and image classification tasks demonstrate that Orchid outperforms attention-based models like BERT and Vision Transformers with smaller model sizes. It also enables processing longer sequences beyond the limitations of dense attention layers.

**Strengths:**

The paper introduces Orchid, a novel deep learning architecture that addresses the quadratic complexity of traditional attention mechanisms while still capturing long-range dependencies and enabling in-context learning. The key innovation is a data-dependent global convolution layer that dynamically adapts its kernel based on the input sequence using a dedicated conditioning neural network. Two simple conditioning networks are designed to maintain shift equivariance in the data-dependent convolution operation. The dynamic convolution kernel allows Orchid to achieve high expressivity with quasilinear scalability for long sequences. Experiments across language modeling and image classification tasks demonstrate that Orchid outperforms attention-based models like BERT and Vision Transformers with smaller model sizes. It also enables processing longer sequences beyond the limitations of dense attention layers. The paper is well-written, with clear descriptions of the Orchid architecture and comprehensive empirical evaluation.

**Weaknesses:**

1. Citation issue: On page 3, there seems to be an issue with the citation at the end of the page. Could the authors please clarify and correct this?
2. Computational complexity: Since the weights in the proposed method come from a neural network (data-driven), wouldn't this increase the computational complexity as the number of layers increases? In general, convolutional blocks do not have data-driven parameters. Could the authors provide more details on how the computational complexity scales with the number of layers?
3. Conditioning networks: To improve the understanding of the paper, could the authors provide more details about the two conditioning networks introduced in the manuscript?
4. Experiments: Which conditioning network is used in the experiments presented in the main paper? Clarifying this would help readers better understand the experimental setup and results.
5. Block-diagonal matrices: The authors mention the use of block-diagonal matrices in the MLP layers for dimension mixing. However, the paper does not provide an ablation study to assess the impact of this design choice on the model's performance and efficiency. Could the authors include such an analysis to justify the use of block-diagonal matrices and provide insights into their role in the overall architecture?
6. Interpretability and explainability: The paper does not provide a detailed analysis of the model's interpretability and explainability. Could the authors develop and discuss techniques to visualize and interpret the learned representations and decision-making process of Orchid? This would help improve the model's transparency and trustworthiness.

Grammatical correction:

- "Moreover, its allows for handling very large sequence lengths that are beyond the limitations of the dense attention layers."

**Questions:**

Please address the questions and concerns provided in the Weaknesses section

**Limitations:**

The authors have talked about various limitations of the paper.

---

> ### Author Response · Authors · 2024-08-07
>
> Thank you for your detailed review and valuable feedback. Your recognition of the contributions of the proposed Orchid architecture and the comprehensive empirical evaluations and also your valuable feedback is appreciated. In the following, we address the points you raised in your review.
>
>
> - *W1: Citation issue at page 3:*
> Thank you for bringing this to our attention. We will ensure that this issue will be fixed.
>
>
> - *W2: Computational Complexity of Conditioning Networks*
> We designed simple conditioning networks using $Conv1d()$ (1D depthwise linear convolution with a short kernel length, typically 3-5) in both the spatial and frequency domains. This architecture choice aims to minimize the number of parameters and computational overhead of the conditioning networks. As a result, the number of parameters does not grow with the sequence length. The computational complexity of $Conv1d()$ is $\mathcal{O}(L D)$, which scales linearly with the sequence length, while FFT computations scale quasilinearly with the sequence length ($\mathcal{O}(L \log L)$).
> It's worth noting that computing the projections for $K$, $Q$, and $V$ in transformers requires $\mathcal{O}(L D^2)$ computation. The runtime benchmarks provided in Appendix C.6 demonstrate the speed performance of Orchid compared to Attention and FlashAttention, highlighting its efficiency.
>
> - *W3: More Details on the Two Conditioning Networks*
> To model the convolution kernels dynamically as a function of the input, we designed two simple conditioning networks that maintain shift equivariance. We chose architectures that minimize parameters and computational overhead while being efficient.
>    - *Conditioning Network I (Equation 2)*: This network first applies a short convolution $Conv1d()$ on the sequence, then transforms the signal into the spectral domain using the Fast Fourier Transform (FFT). Then an absolute value is applied to preserve shift invariance of the kernel.
> Another short convolution $Conv1d()$ is then applied in the spectral domain. This combination of spatial and frequency domain filters effectively captures information from local neighboring tokens and spectral components.
>    - *Conditioning Network II (Equation 3)*: This network first applies two short convolutions $Conv1d()$ on the sequence to obtain two versions of the input sequences, $k'(x)$ and $q'(x)$. Similar to the previous approach, the signal is then transformed into the spectral domain and pointwise multiplied (to implement a fast cross-correlation). Finally, another short convolution $Conv1d()$ is applied in the spectral domain. This approach generalizes Conditioning Network I.
> Both of these conditioning functions are illustrated schematically in Figure 2.1. The output from these conditioning networks is used as the kernel of the long data-dependent convolution that globally mixes the sequence.
>
>
> - *W4: Conditioning Network Used in Experiments*
> We used Conditioning Network I for the experiments presented in the paper, as mentioned in Section C.2 of the Appendix (Experimental Details). Conditioning Network I is slightly more efficient in terms of speed and parameter counts. Based on the ablation study in the appendix, we selected Conditioning Network I for the language, vision, and audio experiments reported in the paper. We will add this clarification to the main body of the paper to ensure readers better understand the experimental setup and results for this selcetion.
>
> - *W5: Impact of using Block-diagonal weights in MLP blocks:*
> Indeed the impact of using M2-MLP blocks can be analyzed by comparing Orchid and M2. In the experiments, we compare Orchid with M2 of similar size, while both are using Block diagonal for MLP, the main difference between M2 and Orchid based architectures is that, in Orchid we deployed the proposed data-dependent long convolution while M2 is following Hyena in using a fixed long convolution. Therefore, the performance improvements observed in Orchid compared to M2 can be attributed to the data-dependent sequence mixing mechanism.
> Using Block diagonal for MLP indeed will result in a sparse implementation of a dense MLP so it reduces the overall number of trainable parameters and reduces computational complexity.

---

### Official Review · Reviewer_nmxo · 2024-07-09

**Soundness:** 3
**Presentation:** 3
**Contribution:** 3
**Rating:** 6
**Confidence:** 3

**Summary:**

This paper introduces the Orchid block, a novel sequence modeling element that employs a convolutional operator with a sample-dependent generated kernel and an $O(n \log n)$ computational complexity with $n$ being a sequence length. The kernel, matching the input sequence length, captures both long- and short-scale dependencies. As would be expected of a similar sequence processing operation, the authors design the kernel generator to be translationally-invariant. The proposed architecture is evaluated on language modeling tasks and image classification (ViT), with experimental results suggesting it outperforms established models like BERT and ViT, as well as recent architectures such as Hyena Hierarchy and Monarch Mixer.

**Strengths:**

1. The paper is well-written and provides clear justification for the core elements of the proposed architecture, including the choice of the sample-dependent, translation-invariant convolution kernel generator.
2. The proposed Orchid block is generally sound (see a question below). The experiments appear to be adequate to evaluate the proposed technique.
3. Empirical results demonstrate that the proposed architecture offers a notable improvement over established Transformer baselines, as well as Hyena Hierarchy and Monarch Mixer models (other recent long convolution-based models). The resulting architecture thus presents a promising advancement.
4. Inclusion of the synthetic in-context learning dataset highlighted an additional significant property of the proposed model: its ability to perform basic associative recall task even with very long sequences.

**Weaknesses:**

1. One weakness limiting a potential significance of this work is that the proposed model cannot be currently applied to causal models. But this can hopefully be addressed in the future work.
2. While the proposed architecture is sub-quadratic and has the capability to efficiently process large inputs, it's crucial to evaluate its performance on real-world very long sequences (not just synthetic toy examples). Current approaches can scale Transformer-based models up to even millions of tokens and it could be quite crucial to demonstrate comparable performance for such long sequences as well.
3. While the design of the translation-invariant generator is sufficiently principled, there are quite a few seemingly arbitrary choices (some nonlinearities, specific function dependencies and more) and a more careful ablation study could be required in the future.

**Questions:**

1. In Equation (3), it is not specified how the nonlinearity $\sigma$ is applied to a _complex_ Fourier spectrum $\mathcal{F}(\dots)$. For the resulting product of $\mathcal{F}^*(\dots) \odot \sigma(\mathcal{F}(\dots))$ to be translation-invariant, it is important that $\sigma$ preserves the phase of the complex number to which it is applied. It makes me conclude that the nonlinearity $\sigma$ acts on the magnitude of the argument, but preserves the phase. Another naive alternative would be to apply $\sigma$ to real and complex parts separately, but this would not preserve the phase and would thus break translational symmetry. Is my understanding correct? And if so, how is $\sigma$ computed in current experiments? If this is correct, I also urge the authors to clarify this point in the publication.
2. There are at least two [Fu et al., 2023] papers being referenced, which creates confusion.

**Limitations:**

The paper does discuss several limitations, of which perhaps the most important one is that it cannot be currently applied to causal models. Authors also highlight the fact that the proposed layer can serve as an alternative to the cross-attention layer. Furthermore, several seemingly arbitrary choices in the model architecture could also be re-evaluated in the future work.

---

> ### Author Response · Authors · 2024-08-07
>
> Thank you for your detailed and thoughtful review of our paper. We appreciate your recognition of the contributions and the strengths of our paper. In the following, we address your points.
>
>
>
> - *W1:*
> Although the current form of the Orchid is not compatible with causal and autoregressive models, since it uses the entire sequence as context, but this property makes it particularly suitable for encoder-based models like BERT and scalable candidate for diffusion models such as [1]. However, it is worth noting that recent studies [2, 3] have raised questions about the optimality of autoregressive models for inference in language models.
>
> - *W2:*
> For sequence of length, we have also evaluated the performance of the proposed method on a raw audio dataset with a large sequence of length 16k in Appendix C.5. While the standard Transformer model could not fit in GPU, Orchid performs on par with the state-of-the-art model on this long sequence modeling task.
>
> - *W3:*
> In addition to the ablation studies presented in Appendix C.2, we have conducted further ablation studies on the conditioning network architecture on associative recall task:
>
>    - Comparison of Local Conv1D Choices: We evaluated different short depthwise linear convolution architectures in the conditioning network, as outlined in Equation (2). This ablation study compared: I) applying Conv1D() in the spatial domain followed by Conv1D() in the frequency domain (as proposed in Equation (2)), II) applying two layers of Conv1D() in the spatial domain only, and III) pplying two layers of Conv1D() in the spectral domain only. The results demonstrated that the proposed method of operating in both spatial and frequency domains (as in Equation (2)) which mixes information from neighboring tokens and spectral token shows the best performance.
>
>    - Impact of Different Nonlinear $\sigma()$ Functions: We evaluated various nonlinear $\sigma()$ functions used in the Type II (cross-correlation) conditioning network (Equation (3)). The nonlinearities tested included \[ Tanh(), Sigmoid(), Softsign(), and Softshrink(), Identity()\], all acting on the magnitude of the argument. The results indicated that dropping this nonlinearity, $\sigma()$ in Equation (3), provides the best performance which is slightly better than nn.Softshrink() and nn.Tanh(). Also, among the nonlinearities those that cross zero perform better.
> Moreover, we also observe that Type II conditioning networks with Identity() and Softshrink() have faster convergence than Type I.
>
> **Questions:**
>
>
> - **Q1: Applying Nonlinearity on the Magnitude in Type II Conditioning Network to Preserve Shift-Equivariance**
> Thank you for your insightful question. As you correctly noted and also mentioned in the paper,
> each of K and Q should remain shift (translation) equivariant, in order that their cross-correlation satisfies the shift-invariance property for the conditioning network. Threfore the nonlinearity $\sigma()$ in Equation (3) should act on the magnitude of the argument while preserving the phase. This approach ensures that each of \( K \) and \( Q \) remains shift-equivariant.
> However, as mentioned in our new ablation study on the associative recall task, we found that dropping this nonlinearity yielded the best performance. We will include this clarification in the final version of the paper.
>
>
> - **Q2: Duplicate References to [Fu et al., 2023]**
> Thank you for bringing this to our attention. We will ensure that this duplicate reference will be fixed.
>
>
> [1] Lou, Aaron, Chenlin Meng, and Stefano Ermon. "Discrete diffusion modeling by estimating the ratios of the data distribution." ICML (2024)
> [2] Nan Ding, Tomer Levinboim, Jialin Wu, Sebastian Goodman, and Radu Soricut. Causal lm is
> not optimal for in-context learning. arXiv preprint arXiv:2308.06912, 2023.
> [3] Gregor Bachmann and Vaishnavh Nagarajan. The pitfalls of next-token prediction. arXiv preprint arXiv:2403.06963, 2024.

---

> > ### Comment · Reviewer_nmxo · 2024-08-10
> >
> > I would like to thank the authors for their detailed response. Additional ablation studies and experimental results with a large (16k) sequence length help alleviate some of my concerns related to method performance. However, the fact that the proposed technique cannot be applied to causal models can still be seen as a limiting factor for its applicability. The authors cited several papers highlighting the limitations of the autoregressive modeling approach, however these types of models still appear to be the golden standard in numerous practical applications. It is also worth mentioning that some of the mentioned papers, are either considering very specific setups (in-context learning setups with the full attention over the context component), or propose to alleviate the limitations of next-token prediction models using modifications within the same causal setup. Considering concerns raised by other reviewers and the authors replies, I would like to keep my current favorable score.

---

### Official Review · Reviewer_D4Wg · 2024-07-15

**Soundness:** 3
**Presentation:** 2
**Contribution:** 4
**Rating:** 7
**Confidence:** 4

**Summary:**

This paper presents Orchid, a novel method that conditions long convolutional layers based on the input, to obtain global context and in-context learning abilities. This is achieved through the use of a conditioning network that acts both on the spatial and frequency domain to mix close spatial and spectral tokens, respectively. The resulting model surpasses other subquadratic models like Hyena and M2. It’s empirical results are compelling demonstrating the abilities of the proposed model over BERT and ViT-like models.

**Strengths:**

- The idea for input-dependence of convolutional kernels presented in the paper is novel, sound and very appealing.
- The empirical evidence shows compelling evidence of the proposed model abilities.

**Weaknesses:**

- In my understanding, I am afraid that certain parts of the proposed model oversell what the model is capable of. Specifically, in Line 125, the authors argue that "This allows each input token to attend to the entire sequence with personalized, adaptive weights derived from its specific representation".  However, to the best of my understanding, Orchid only considers local information --both spatial and spectral-- for conditioning. See also Line 302-304 and 308. The authors should be clear and fair regarding the abilities of the proposed model.

- i am afraid I do not truly understand the Orchid block. Note that Figure 2.1 has an MLP block at the beginning of the block, which is then completely ignored both in the text and in the reference implementation. I would strongly encourage the authors to clearly state how the Orchid block works in practice.

- Although encouraging, I feel that the empirical section misses many important components that would improve the impact and adoption of Orchid.
  - First, the associative recall tasks are only compared with other long conv models, some of which have been previously shown not to work in such tasks, e.g., H3, CKConv. For Orchid to be adopted in practice, I think that it would be very important to compare both to existing Transformer-like architectures as well as exiting input-dependent SSMs, e.g., Mamba.
  - Related to the previous comment, Mamba has shown that existing long conv models do not perform well in the selective copying task. How does Orchid perform in this task?
  - On the BERT experiments, the authors should include ablation studies to study the impact of using M2-MLP blocks as opposed to normal ones. It is not clear how much of a benefit (or decrease) results from this replacement, which is not inherent to Orchid.
  - On the image classification experiments, the authors compare with methods that perform very poorly, e.g., 84% acc on CIFAR-10. I would encourage the authors to compare to more relevant existing methods.
  - Related to the previous comment, one of the main advantages of long-conv models and Orchid is the fact that patches can become very small. If the accuracy results from the previous comment are limited by patchification, it would be very interesting to explore the performance differences when this module is removed.
  - Related to the previous comment, the same experiment would be very valuable for ImageNet1k. Recent papers have shown that the smallest the patch the better the accuracy. Showing that Orchid scales well to ImageNet in this setting would be very appealing and would also potentially increase the impact of the paper.
  - Still on the image processing setting, as far as I understand, it is very easy (not to say trivial) to expend Orchid to 2D data. Is there any reason as of why the authors did not use 2D orchid blocks for the image tasks?

**Questions:**

- Line 27. Please add references.

- In the appendix C.2, the authors state that Type-I conditioning combined with DTC seems to work best. Is this truly the case? If so, then I do not understand why other components are introduced that are not used at the end. Note that Type I is much simpler than Type II, both conceptually, in terms of implementation and speed. If this is the case, then that space can also be rather used for things that are used in the experimental section.

- The authors also introduce data dependent convolution as an alternative to Cross-attention. But then, never use it. Again, this space can be better used to outline the components that are used in the experiments, or to extend the experimental section of the method.

**Limitations:**

The authors clearly state the limitations of the proposed method.

### Conclusion

Whilst I very much like the idea proposed here, and acknowledge its novelty and potential impact, I am not sure that in its current for this paper would be as impactful as I believe it could be. Therefore, I am hesitant to support acceptance. Note that I strongly believe that this paper could be very impactful. However, I believe that multiple adjustments must be made. With that being said, I am happy to increase my score should my concerns and comments be addressed.

---

> ### Author Rebuttal · Authors · 2024-08-07
>
> Thank you for your constructive feedback on our paper. We appreciate your recognition of the novelty and soundness of our approach, as well as the compelling empirical evidence. In the following, we address the points you raised in your review.
>
> -  *Orchid is using local info for spectral and spatial:*
> In the proposed method, the long data-dependent convolution has an adaptive kernel that spans the entire length of the input sequence. This means that the convolution operation performed by Orchid indeed mixes the entire sequence globally using the proposed adaptive input dependent kernels, rather than being restricted to local windows.
> While the conditioning network’s input are local windows in both the spectral and spatial domains to determine the adaptive kernel, the convolution itself applies this kernel across the entire sequence. Thus, each input token is effectively mixed (convolved) with the entire sequence,
>
> -  *The role of MLPs at the beginning of the block:*
> The MLPs included at the beginning of the Orchid block serve as feature (dimension) mixing components that are commonly used in various sequence modeling architectures, such as Hyena, M2, and SSM. In practice, these MLPs perform a pointwise mixing of features, which is a standard technique employed both before and after sequence modeling operations in architectures like transformers.
> Since the main focus of our paper is on sequence modeling using the data-dependent global convolution mechanism the text focused on these parts in detail. However, we will clarify the role of these MLPs in the final version.
> -  *Impact of using M2-MLP blocks on the final results:*
> Indeed the impact of using M2-MLP blocks can be analyzed by comparing Orchid and M2. In the experiments we compare Orchid with M2 of similar size, while both are using Block diagonal for MLP, the main difference between M2 and Orchid based architectures is that, in Orchid we deployed the proposed data-dependent long convolution while M2 is following Hyena in using a fixed long convolution. Therefore, the performance improvements observed in Orchid compared to M2 are attributed to the data-dependent sequence mixing mechanism
> Using Block diagonal for MLP indeed will result in a sparse implementation of a dense MLP so it reduces the overall number of trainable parameters and reduces computational complexity
>
> **Questions:**
>
> -  *Q1: reference for line 27*
> Thank you for pointing this out.  We will add the references to support the statements such as [1]
>
>
> -  *Q2: Why Type II is introduced in the text?*
> The Type II (cross-correlation) approach generalizes the Type I (magnitude-based) approach and encompasses it as a special case. The introduction of Type II was intended to provide a more comprehensive view of the conditioning techniques available.
> Moreover, recent ablation studies, detailed in the rebuttal PDF, indicate that Type II combined with DCT performs better than Type I with DCT.
> However, since it is slightly more efficient in speed and parameter counts, Type I with DCT was selected for the in language, vision and audio experiments reported in our paper
>
> -  *Q3: Cross-attention functionality of Orchid:*
> In the text we emphasize the novel capabilities of the Orchid model.
> The other long convolution based methods, such as Hyena and M2, and SSM-based methods such as Mamba, are not inherently applicable as an alternative to Cross-attention, so in this work we emphasize that the Orchid model is not only input-dependent but also its kernel can be conditioned on the an arbitrary input of any length so we call it data-dependent to infer more general than input-dependent modules. Implementing this ability of the proposed model could be a valuable direction for future work.
>
>
> [1] Nguyen, Eric, et al. "Hyenadna: Long-range genomic sequence modeling at single nucleotide resolution." Advances in neural information processing systems 36 (2024).

---

> ### Author Response · Authors · 2024-08-12
>
> - *Associative Recall on SSM-Based Models:*
> In the following table, we compare the performance of Orchid against other long convolution models and an input-dependent SSM (Mamba) on the associative recall task. As the results indicate, Orchid achieves state-of-the-art performance among existing scalable sequence models for this task.
>
> **Table AR2:** This table presents the test accuracy (in %) of in-context learning on the associative recall task with a sequence length of 128 and varying vocabulary sizes.
>
> | Vocabulary Size | 20   | 30	| 40	|
> |-----------------|------|-------|-------|
> | **Transformer** | 100  | 100   | 100   |
> | **CKConv**  	| 91   | 25.7  | 20.4  |
> | **H3**      	| 71.5 | 13.2  | 10.2  |
> | **Hyena**   	| 93   | 38.8  | 12.4  |
> | **Mamba**   	| 100  | 100   | 35.8  |
> | **Orchid**  	| 100  | 99.4   | 99.2  |
>
> - *Expanding Orchid to 2D long convolution long:*
> The Orchid block, with its input-dependent long convolution, local depthwise linear convolution (Conv1d), and element-wise multiplications, is inherently extendable to multi-dimensional data.
> However, our primary focus in this work was on designing an efficient and scalable architecture specifically for *sequence modeling*. Expanding our architecture to include 2D-convolutional long-range approaches, while valuable, was beyond the scope of our current study and is an interesting future work.
>
>
> **UPDATE**
> - We appreciate your suggestion and have included additional results in our "official comment" titled [*Updated Results on CIFAR-10*](https://openreview.net/forum?id=a75F45dBHK&noteId=h2L7DEaIKv), where Orchid is compared against other high-performance models. As the results show, using smaller image patches boosts the performance. Additionally, we include results for Orchid with the Type II conditioning network, which offers a slight improvement in accuracy on CIFAR-10.

---

> > ### Comment · Reviewer_D4Wg · 2024-08-13
> >
> > Dear authors,
> >
> > Thank you so much for your response.
> >
> > My questions regarding the experimental section have been answered. However, multiple questions that remain open / unanswered:
> >
> > *While the conditioning network’s input are local windows in both the spectral and spatial domains to determine the adaptive kernel, the convolution itself applies this kernel across the entire sequence. Thus, each input token is effectively mixed (convolved) with the entire sequence*
> >
> > +-> It is clear to me that the input is mixed across the entire sequence. However, as indicated before, the conditioning is local. This should be made clear in the paper as this elucidates improvements that can be done to the method in the future.
> >
> > Regarding **questions**:
> >
> > * Respectfully, I am strongly against that kind of "flag planting" that is made in the paper for both the Type II conditioning as well as for the Cross-Attention alternative. I believe that only things that are used in the paper *should* be included. I believe that this hinders progress on the field, as other researchers --or even yourself-- might feel that contributing and experimenting with those parts are not worth pursuing, as it has already "been done before" (which is not the case). I would strongly encourage the authors to remove the parts that are not used in the model.
> >
> > * There is no rebuttal PDF attached anywhere.
> >
> > * The authors did not answer my comment regarding Figure 2.1. I would encourage the authors to improve this image as it is the main image of the paper. It would aid clarity to use the conventional way of illustrating network blocks, as has been done in several papers before, eg., ResNet, Transformer, etc.
> >
> > Due to the observations outlined before, I am not compelled to support a clear acceptance. I therefore will maintain my score.
> >
> >
> >
> >
> >
> >
> >
> >
> >
> >
> > Unfortunately, I am unsatisfied by your rebuttal response.

---

> > > ### Author Response · Authors · 2024-08-14
> > >
> > > Dear Reviewer,
> > >
> > >
> > > Thank you for your response. Regarding your questions:
> > >
> > > -  *Local conditioning network on spatial and spectral:*
> > >    - To address your question, let's clarify this with an analogy. In the conditioning network of Orchid, the pair of signals $ k(x) $ and $ q(x) $ are analogous to the key $ k(x) $ and query $ q(x) $ in the attention mechanism. In both Orchid and the attention mechanism, these components are modeled using local neural networks: attention utilizes pointwise linear projections, while Orchid employs local $ Conv1D $ operations in both the spatial and spectral domains.
> > > Once $ k(x) $ and $ q(x) $ are computed, attention mechanisms calculate the input-dependent attention score matrix $ A(x) $ and subsequently compute the output as $ y = A(x) v $. In contrast, Orchid’s conditioning network performs a cross-correlation— a global operation—between $ k(x) $ and $ q(x) $ to derive the convolution kernel $ h(x) $, followed by global convolution $ y = h(x) * v $.
> > > Thus, while the inner blocks (the input-dependent networks that compute the convolution kernel in Orchid or the attention matrix in attention mechanisms) operate locally on the inputs, the outer blocks (such as the matrix product in attention, cross-correlation or convolution in Orchid) perform global sequence mixing. This approach ensures fixed (or sublinear) parameter scaling with respect to sequence length, preventing the model size from growing excessively with sequence length.
> > > This analogy also applies to Type I, as it is a special case of Type II where $k(x)=q(x)$.
> > >
> > >    - Although it is mentioned in line 148-150 that the conditioning network acts on both local spatial and spectral components, we will add this discussion in the final version to prevent potential misunderstandings.
> > >
> > > -  *Experimental results on Type II:*
> > > By including discussions on Type II conditioning  and the Cross-Attention alternative we aimed to highlight and clarify the distinct features of the data-dependent global convolution mechanism in Orchid. We believe that these discussions provide valuable insights that help better understand the architectural design of Orchid and also foster further exploration and innovation in the field by adopting it in other applications by the researchers.
> > > Moreover, we have conducted experimental evaluations using Orchid with the Type II conditioning network. The results, which are shared in the "official comment" titled *Updated Results on CIFAR-10*, demonstrate a slight improvement in accuracy on CIFAR-10, supporting the potential benefits of this approach.
> > >
> > > -  *Model Architecture in Figure 2.1:*
> > > The paragraph in our rebuttal starting with “The role of MLPs at the beginning of the block” was intended to address your concerns about this figure. The MLPs at the beginning of the Orchid block are indeed pointwise feature mixing components, a design choice commonly seen in various architectures such as Hyena, M2, and SSM. However, the primary focus of our paper is on sequence modeling using the data-dependent global convolution mechanism.
> > > The current illustration style in Figure 2.1 was chosen to best represent the distinct features of the Orchid block, particularly highlighting its global convolution mechanism. Such style has been adopted in other relevant models, such as M2 and Mamba which we believe is suitable for our purposes.
> > >
> > >
> > > Thank you once again for your valuable feedback, and we hope our response has addressed all your questions.

---

> > > > ### Comment · Reviewer_D4Wg · 2024-08-14
> > > >
> > > > Dear authors,
> > > >
> > > > Thank you for your answer.
> > > >
> > > > **Local conditioning network on spatial and spectral.** That is a good analogy. Thank you for the clarification. It would perhaps be nice to add a section on the appendix explaining this connection. This will probably help readers with doubts about that conditioning part --like myself--, get a better understanding of its relation to self attention.
> > > >
> > > > **Model Architecture in Figure 2.1.** I understand. Could you please add an additional image to the Appendix following those traditional standards to complement the paper? I feel that this still would add much clarify to the paper.
> > > >
> > > > Under the understanding that the authors will add these two parts to the paper I am happy to increase my score. It is now 7.

---

### Official Review · Reviewer_pFWN · 2024-07-17

**Soundness:** 3
**Presentation:** 3
**Contribution:** 3
**Rating:** 6
**Confidence:** 4

**Summary:**

Authors introduce a method for addressing the quadratic computational complexity of the attention mechanism while retaining expressivity and model performance from transformer models. Whereas previous approaches have achieved sub-quadratic computational efficiency - e.g. hyena, ssms and CKConv - authors argue (and show in their experiments) that these approaches limit model expressivity and performance - specifically in in-context learning settings. To this end, authors propose a novel long-range convolutional layer that is data-dependent, i.e. conditioned on the input. Orchid retains shift-equivariance properties of conventional convolution-based architectures, while being more expressive. Authors show superior performance over previous approaches in different domains (text and image data).

**Strengths:**

- Authors introduce an innovative method for increasing the expressivity of subquadratic methods for long-range dependencies.
- The paper is well-written, authors explain and motivate their modelling choices well, and provide helpful figures.
- The approach of conditioning convolutional kernels based on input data is interesting in its own right and might warrant exploration in architectures not specifically tailored for modelling long-range dependencies.

**Weaknesses:**

- Limited set of experiments and comparisons against baselines. Although authors show results also on image data, they do not compare against 2D-convolutional long-range approaches which flimits interpretability of the results.
- Authors do not thoroughly explore their shift-invariance constraints, which might not be appropriate in all settings, i.e. I can imagine that for textual data absolute positioning in a sentence does impact semantic meaning. On the other hand, authors provide good motivation for their choice of shift-invariance.

**Questions:**

- Regarding the synthetic in-context learning task, I might be misunderstanding something, but why are shorter sequences more challenging in this task (i.e. Hyena/ CKConv/H3 achieve lower accuracy on shorter sequences)? Isn't it generally more challenging to capture longer-range dependencies?
- Why do you choose not to compare against CKConv [1] or a later improvement CCNN [2] in the image classification task (these models ? These methods achieve substantially better results on CIFAR, making me question the validity of your claims regarding the advantage of your model in a broad range of application domains.

[1] Romero, D. W., Kuzina, A., Bekkers, E. J., Tomczak, J. M., & Hoogendoorn, M. (2021). Ckconv: Continuous kernel convolution for sequential data. arXiv preprint arXiv:2102.02611.
[2] Knigge, D. M., Romero, D. W., Gu, A., Gavves, E., Bekkers, E. J., Tomczak, J. M., ... & Sonke, J. J. (2023). Modelling Long Range Dependencies in $ N $ D: From Task-Specific to a General Purpose CNN. arXiv preprint arXiv:2301.10540.

**Limitations:**

Authors sufficiently discussion limitations of their approach.

---

> ### Author Rebuttal · Authors · 2024-08-07
>
> Thank you for your thoughtful review and valuable feedback. We appreciate your recognition of the contributions and the strengths of our paper. In the following, the points you raised in your review are addressed.
>
> - *W1: Compare against 2D-convolutional long-range approaches:*
> The Orchid block, with its input-dependent long convolution, local depthwise linear convolution (Conv1d), and element-wise multiplications, is inherently extendable to multi-dimensional data.
> However, our primary focus in this work was on designing an efficient and scalable architecture specifically for sequence modeling. Expanding our comparisons to include 2D-convolutional long-range approaches, while valuable, was beyond the scope of our current study and is an interesting future work.
> Regarding your point on the limited set of experiments against baselines, Appendix C5 also explores the ability of Orchid to learn long-range dependencies in speech classification tasks with long sequences. In these experiments, we compared Orchid against CKConv, Performer, and SSM-based models.
>
>
> - *W2: Absolute positioning in a sentence might impact semantic meaning:*
> Our approach to shift invariance focuses on preserving the relational positions of tokens. While changing the order of the tokens/words and their absolute positioning might impact the semantic meaning, we expect that if an entire sentence is shifted and pad tokens are appended before or after it,  the sequence retains its semantic meaning. Indeed the relational positions of the tokens remain consistent in such cases.
> Furthermore, the positional embeddings added to the token embeddings, at the beginning of language models, encode the absolute positions of the tokens, enabling the model to generate semantic differences when the order is changed. Moreover, ​​to achieve a location-dependent filtering scheme, we complement the data-dependent convolution with element-wise multiplications which allows the model to emphasize specific tokens in a sequence.
>
> - *Q1: Why are shorter sequences more challenging in the in-context learning task?*
> Shorter sequences pose a unique challenge in in-context learning tasks because specific (key, value) pairs appear less frequently within the string. This reduced frequency means the model has fewer opportunities to learn and generalize these associations. Additionally, when the vocabulary size also increases, the task becomes even more challenging for the model. The combination of infrequent pair repetitions and a larger vocabulary require a more expressive model architecture to effectively capture and utilize these (key, value) pairs within shorter sequences.
>
> - *Q2: Compare against CKConv variants:*
> Fixed long convolution kernel term (bias term) in Orchid and also convolution kernel in Hyena and M2 model are built upon long convolution kernel in CKConv, so by comparing against Hyena and M2 we explore the impact of the proposed input-dependent convolution that was introduced by Orchid model.
> Moreover,  in speech classification tasks with raw speech, Orchid is compared against CKConv.

---

> > ### Comment · Reviewer_pFWN · 2024-08-09
> >
> > I thank the authors for their response. Most my concerns have been answered, except for the seemingly arbitrary choice by the authors to compare against ckconv in one experiment but not in another experiment where it is outperformed by CKConv. I would recommend adding these results.

---

> > > ### Author Response · Authors · 2024-08-13
> > >
> > > Dear Reviewer,
> > >
> > > Thank you for your valuable feedback. We appreciate your suggestion and have included additional results in our "official comment" titled *Updated Results on CIFAR-10*, where Orchid is compared against other models, including CKConv and CCNN. In the original submission, our primary focus was on comparing Orchid with Vision Transformer (ViT) baseline models.
> > >
> > > We hope our response has addressed all your questions.

---

### Author Response · Authors · 2024-08-13
**Updated Results on CIFAR-10**

Dear Reviewers,

In response to the feedback, we are including additional experimental results on the CIFAR-10 dataset, complementing the results presented in the paper. For these experiments, the images were converted into sequences using patches of sizes $(4 \times 4)$, $(2 \times 2)$, or $(1 \times 1)$ pixels. We designed Orchid-based models using a ViT architecture composed of 6 Transformer layers, with hidden sizes of either 128 and 220 for Orchid-ViT-s and Orchid-ViT-m, respectively.

**Table RBTL1:** Test performance on the CIFAR-10 dataset flattened with patch sizes $(4 \times 4) $, $(2 \times 2) $, and $(1 \times 1 )$ (sCIFAR). Orchid-ViT-s (Orchid-ViT-m) refers to the ViT architecture composed of 6 layers with hidden sizes of 128 (220). Baseline results are referenced from [1] and [2]. For this result on Orchid-ViT-s  (4 \times 4), we tuned base learning rate over (1e-3, 5e-3, 1e-2).
| **Model (Patch Size)** | **Size** | **Accuracy (%)** |
|------------------------|----------|------------------|
| **ViT** $(4 \times 4)$  | 1.2M	| 78.6         	|
| **ViT+Monarch** $(4 \times 4)$ | 607K	| 79.0         	|
| **Hyena-ViT** $(4 \times 4)$  | 1.3M	| 80.6         	|
| **M2-ViT** $(4 \times 4)$  | 741K	| 80.8         	|
| **Orchid-ViT-s** $(4 \times 4)$  | 735K	| 88.5    	|
|------------------------|----------|------------------|
| **CKConv** $(1 \times 1)$  | 1M  	| 63.74        	|
| **M2-ViT** $(1 \times 1)$  | 741K	| 91.0         	|
| **S4** $(1 \times 1)$  | 7.8M	| 91.13        	|
| **CCNN** $(1 \times 1)$  | 2M  	| 93.08        	|
|------------------------|----------|------------------|
| **Orchid-ViT-s** $(2 \times 2)$  | 790K	| 92.2         	|
| **Orchid-ViT-s  Type II** $(2 \times 2)$ | 799K	| 92.3         	|
| **Orchid-ViT-m** $(2 \times 2)$  | 2.1M	| **93.33**    	|
| **Orchid-ViT-m** $(1 \times 1)$  | 2 M	| 93.0   	|

**Key observations from this empirical study:**
- Orchid outperforms baseline models on the CIFAR-10 dataset.
- Using smaller image patches can boost the performance.
- The Type II conditioning network offers a slight improvement in accuracy on Orchid-ViT-s.

[1] Daniel Y Fu, et al. Monarch mixer: A simple sub-quadratic GEMM-based architecture. arXiv preprint arXiv:2310.12109, 2023.
[2] Knigge, D. M., et al. Modelling Long Range Dependencies in D: From Task-Specific to a General Purpose CNN. arXiv preprint arXiv:2301.10540, 2023.

---

### Decision · Program_Chairs · 2024-09-25

**Decision:**

Accept (poster)

**Comment:**

The paper introduces Orchid, a novel approach to enhancing the expressivity of subquadratic methods for long-range dependencies through a data-dependent global convolution mechanism. The reviewers generally agree that the paper offers a significant contribution to the field of sequence modeling, with potential applications across various domains such as text and image data (pFWN, nmxo, eXPv). The strengths of the paper lie in its clear justification of design choices, soundness of the proposed architecture, and empirical results that demonstrate improvements over established models like BERT, ViT, and recent architectures like Hyena Hierarchy and Monarch Mixer.

However, some concerns were raised by the reviewers that should be addressed. One notable issue is the “flag planting” concern highlighted by D4Wg, where the paper introduces elements such as Type II conditioning and cross-attention alternatives without fully integrating them into the main experimental validation. While the authors have provided additional experimental results and clarifications, it is important that the final manuscript clearly distinguishes between fully validated contributions and ideas that have not been extensively tested. Other concerns include the need for more thorough comparisons with baselines such as CKConv (pFWN) and further validation on practical datasets beyond synthetic data (nmxo).

Despite these concerns, I believe the novelty and potential impact of the proposed method outweigh the issues raised. I recommend accepting the paper, with a strong suggestion to the authors to revise the manuscript to clearly delineate between validated contributions and those that require further exploration. This will help ensure that the paper accurately reflects the state of its contributions and supports future research in the field.